# Intrabody-Induced Cell Death by Targeting the *T. brucei* Cytoskeletal Protein *Tb*BILBO1

Christine E. Broster Reix,[a] Miharisoa Rijatiana Ramanantsalama,[a] Carmelo Di Primo,[b] Laëtitia Minder,[b,c] Mélanie Bonhivers,[a] Denis Dacheux,[a,d] Derrick R. Robinson[a]

[a]University of Bordeaux, CNRS, Microbiologie Fondamentale et Pathogénicité, UMR 5234, Bordeaux, France
[b]University of Bordeaux, INSERM U1212-CNRS UMR 5320, ARNA laboratory, European Institute of Chemistry and Biology, Pessac cedex, France
[c]University of Bordeaux, CNRS UMS3033-INSERM US001, European Institute of Chemistry and Biology, Pessac cedex, France
[d]Bordeaux INP, Microbiologie Fondamentale et Pathogénicité, UMR 5234, Bordeaux, France

Christine E. Broster Reix and Miharisoa Rijatiana Ramanantsalama contributed equally to this article. Author order was determined alphabetically.

**ABSTRACT** *Trypanosoma brucei* belongs to a genus of protists that cause life-threatening and economically important diseases of human and animal populations in Sub-Saharan Africa. *T. brucei* cells are covered in surface glycoproteins, some of which are used to escape the host immune system. Exo-/endocytotic trafficking of these and other molecules occurs via a single copy organelle called the flagellar pocket (FP). The FP is maintained and enclosed around the flagellum by the flagellar pocket collar (FPC). To date, the most important cytoskeletal component of the FPC is an essential calcium-binding, polymer-forming protein called *Tb*BILBO1. In searching for novel tools to study this protein, we raised nanobodies (Nb) against purified, full-length *Tb*BILBO1. Nanobodies were selected according to their binding properties to *Tb*BILBO1, tested as immunofluorescence tools, and expressed as intrabodies (INb). One of them, Nb48, proved to be the most robust nanobody and intrabody. We further demonstrate that inducible, cytoplasmic expression of INb48 was lethal to these parasites, producing abnormal phenotypes resembling those of *Tb*BILBO1 RNA interference (RNAi) knockdown. Our results validate the feasibility of generating functional single-domain antibody-derived intrabodies to target trypanosome cytoskeleton proteins.

**IMPORTANCE** *Trypanosoma brucei* belongs to a group of important zoonotic parasites. We investigated how these organisms develop their cytoskeleton (the internal skeleton that controls cell shape) and focused on an essential protein (BILBO1) first described in *T. brucei*. To develop our analysis, we used purified BILBO1 protein to immunize an alpaca to make nanobodies (Nb). Nanobodies are derived from the antigen-binding portion of a novel antibody type found only in the camel and shark families of animals. Anti-BILBO1 nanobodies were obtained, and their encoding genes were inducibly expressed within the cytoplasm of *T. brucei* as intrabodies (INb). Importantly, INb48 expression rapidly killed parasites producing phenotypes normally observed after RNA knockdown, providing clear proof of principle. The importance of this study is derived from this novel approach, which can be used to study neglected and emerging pathogens as well as new model organisms, especially those that do not have the RNAi system.

**KEYWORDS** BILBO1, cytoskeleton, intrabody, nanobody, parasite, trypanosoma

Trypanosomes are flagellated protists comprised of pathogenic species capable of infecting a wide range of vertebrate hosts, including humans and domestic and wild animals. Trypanosomes cause lethal and economically important human and animal diseases worldwide (1, 2). A limited number of diagnostic tests and treatments are available for human and animal African trypanosomiasis. Fortunately, the first oral trypanocidal drug, fexinidazole,

Address correspondence to Derrick R. Robinson, derrick-roy.robinson@u-bordeaux.fr.

was recently approved for use in humans in Africa, which simplifies the treatment regimen for certain cases (3, 4). However, millions of wild animals and domesticated livestock are still at risk of infection, while the economic burden of the disease to the African economy is estimated to be 4.7 billion U.S. dollars (USD) per year due to the death of around 3 million head of cattle (5) (http://www.fao.org/paat/the-programme/the-disease/en/). Understanding the basic biology of any virulent organism is important to fully understand its pathogenicity. Furthermore, new targets, tools, and models are needed.

Trypanosomes possess a highly organized cytoskeleton comprised primarily of microtubules but also many cytoskeleton-associated proteins (5–12). Most pathogenic trypanosomes possess a single copy organelle called the flagellar pocket (FP) at the site where the flagellum exits the cell body, which is physically connected to the cytoskeleton (12–17). In many species, the FP is the sole site of exo- and endocytosis. In *Trypanosoma brucei*, the FP is essential for survival and has a cytoskeleton component called the flagellar pocket collar (FPC) that circumvents the neck of the FP at the site where the flagellum exits the cell (13, 14, 18, 19). The FPC forms a cytoskeletal boundary or interface at the site of contact between the pellicular, flagellar, and FP membranes (13, 20, 21). To date, only a few FPC and FPC-associated proteins have been identified and only two essential proteins have been characterized. The first is *Tb*BILBO1 (67 kDa; 587 amino acids [aa]), which is the main protein component of the flagellar pocket collar (FPC), and the second is *Tb*MORN1 (40.9 kDa; 358 aa), which is present in the hook complex (HC), a structure intimately linked and immediately distal to the FPC, and is thought to indirectly play a role in facilitating protein entry into the cell. Two other FPC and FPC-associated proteins, *Tb*BILBO2 and *Tb*FPC4, respectively, are *Tb*BILBO1 binding partners involved in a tripartite interaction (21, 22).

The N terminus of *Tb*BILBO1 adopts a ubiquitin-like fold, followed by two calcium-binding EF-hand domains (EFh1: amino acid residues 185 to 213 and EFh2 221 to 249). These domains are followed by a large coiled-coil domain (CC; amino acids 263 to 566) and a C-terminal leucine zipper (LZ; amino acid residues from 534 to 578) that are involved in dimerization and polymerization, respectively (20, 22–25). RNA interference (RNAi)-mediated knockdown of *Tb*BILBO1 in the cultured, insect procyclic form (PCF) parasites prevents FPC and FP biogenesis, induces the aberrant repositioning of the new flagellum (detached from the cell body along its length), and results in parasite cell death. Knockdown of *Tb*BILBO1 in the mammalian bloodstream form (BSF) is rapidly lethal (13). The FPC is also intimately associated with the microtubule quartet (MtQ), a subset of specialized microtubules that originate near the basal bodies of the flagellum and grow very close to, if not through, the FPC (22, 26, 27). Previous published work has identified binding of *Tb*MORN1 and *Tb*SPEF1 (a microtubule binding and bundling protein) onto detergent and salt-extracted MtQ, and prior to those studies, our anti-*Tb*BILBO1 (IgM) antibody demonstrated both an FPC and a minor MtQ signal (13, 27, 28).

We searched for novel, easily modifiable molecular and immunological tools to study BILBO1 and decided to raise nanobodies against *Tb*BILBO1. Nanobodies are derived from a specific class of heavy chain only antibodies (HCAb) found naturally in Camelidae (alpacas, camels, llamas, and vicunas) and also in Elasmobranchs (cartilaginous fish: sharks and rays) (29, 30). Nanobodies are recombinant antibody fragments derived from the antigen-binding domain (variable domain, VHH) of heavy chain antibodies and are much smaller than traditional antibodies, being ~15 kDa in mass (31–33). The use of nanobodies is greatly expanding in cellular and molecular biology, diagnostics, and medicine. Indeed, in 2018, caplacizumab was approved as the first nanobody-based drug (34). Intrabodies (INbs), cytoplasmic nanobodies, have many potential roles ranging from pathogen detection to killing tumor cells, as well as therapeutic use (35–38). Numerous studies have used nanobodies to characterize a wide variety of proteins, but to date, only one nanobody has been developed against a trypanosome cytoskeleton protein, (Nb392), targeting the paraflagellar rod protein (PFR1) (39).

RNA interference (RNAi) has been used to greatly enhance knowledge of gene and consequently protein function in living cells. However, RNAi can have several limitations, such as incomplete knockdown of the target protein. Furthermore, the double-stranded

RNA produced during RNAi has a relatively short half-life compared to that of an intrabody. Intrabodies are nanobodies expressed cytoplasmically in the target cell and have the benefit of having high antigen specificity, the ability to recognize and bind to a structural epitope not accessible by conventional antibodies, and potential as tools for many biological systems (40–43). One advantage of intrabodies is that they act at a posttranslation stage and therefore can target a specific isoform or posttranslational modification of a protein (43). Also, in the case of essential or structural proteins, induction of the expression of INbs at a specific cell cycle, life cycle stage, or point in organelle biogenesis may provide very subtle and precise phenotypes. In this regard, nanobodies also have advantages over conventional antibodies due to their smaller size, allowing access to cryptic epitopes and ability to cross the blood-brain barrier (44, 45). Conventional antibodies have a further disadvantage of multi-epitope cross-reaction, a phenomenon exemplified by IgE in allergen cross-reaction, whereas nanobodies are, theoretically, single epitope specific (46, 47).

We show here that our anti-*Tb*BILBO1 (aa 1-110) rabbit polyclonal, purified Nb48, and intrabody 48 (INb48) all bind to the FPC and the MtQ. Importantly, we demonstrate that intrabodies raised against *Tb*BILBO1 can have effects comparable to RNAi knockdown and we establish that anti-*Tb*BILBO1 intrabody (INb48) identifies its target protein when expressed in a homologous mammalian system *in vitro* and within the cytoplasm of *T. brucei*. Our data illustrate, for the first time, a functional anti-cytoskeletal intrabody in *T. brucei* that is able to target and bind precisely to its target protein epitope and block biogenesis of a cytoskeletal structure leading to rapid cell death. These data support the hypothesis that targeting minor, yet essential, cytoskeletal proteins is of considerable merit in the search to understand parasite biology. The results also suggest that the use of intrabodies might be useful in organisms that do not have RNAi machinery or to characterize the cell biology of emerging pathogens, marine organisms, protists, and the numerous new model organisms such as those described by Faktorová et al. (48).

## RESULTS

**Production of anti-*Tb*BILBO1 nanobodies.** Full-length histidine-tagged *Tb*BILBO1 protein was expressed in bacteria and then used to immunize an alpaca to produce *Tb*BILBO1-targeting nanobodies by the Nanobody Service Facility (NSF, VIB, Brussels). Seven different nanobodies (Nbs) were identified following phage display screening. These were assigned to three different groups based on their amino acid sequences. It was hypothesized that Nbs in the same group would recognize the same epitope, but their other characteristics, such as affinity, potency, stability, and expression yield, could be different. We chose to focus on one nanobody from each group for our current studies: Nb48, Nb9, and Nb73.

**Nb48 as a functional tool for immunolabelling *Tb*BILBO1.** The Nbs were cloned in the pHENC6c vector in frame with a hemagglutinin (HA) and histidine tag (Nb$_{::HA::6His}$), allowing the secretion of the Nbs in the periplasm, their purification using metal affinity chromatography, and their detection (Fig. S1). Samples of noninduced ($-$) and induced bacteria ($+$), pellet (P), and periplasmic extracts (Ex) were probed by Western blotting with anti-HA (Fig. S2A and B). For simplicity, the HA::6His-tagged nanobodies are hereafter called Nb48, Nb9, and Nb73 (with tags written in subscript). We tested purified Nb48$_{::HA::6His}$, Nb9$_{::HA::6His}$, and Nb73$_{::HA::6His}$ by using anti-HA for detection in immunofluorescence microscopy (IFA) on *T. brucei* wild-type (WT) procyclics (PCF) detergent-extracted cells (cytoskeletons) (Fig. 1). For convenience, in Figures 1 and 2, the cells that are probed with purified nanobodies are labeled "Anti-Nb," plus the relevant nanobody name. In these experiments, purified Nb48$_{::HA::6His}$ labels endogenous *Tb*BILBO1 at the FPC, as seen by colocalization with anti-*Tb*BILBO1 (polyclonal antibody 1-110) (Fig. 1A). Nb9$_{::HA::6His}$ does not label wild-type cytoskeletons expressing endogenous *Tb*BILBO1 protein (Fig. 1B) but does label *Tb*BILBO1 in cells that overexpress *Tb*BILBO1$_{::3cMyc}$ (Fig. 1C). Importantly, Nb73$_{::HA::6His}$ was negative in our experiments and did not label *Tb*BILBO1 in either endogenous (Fig. 1D) or overexpressed levels of *Tb*BILBO1 (not shown). To exclude the possibility of cross-reaction between primary and/ or secondary antibodies in double labeling protocols, Nb48$_{::HA::6His}$ and anti-*Tb*BILBO1 (1-110) were also used alone to probe cytoskeletons. In both cases, the subsequent labeling was

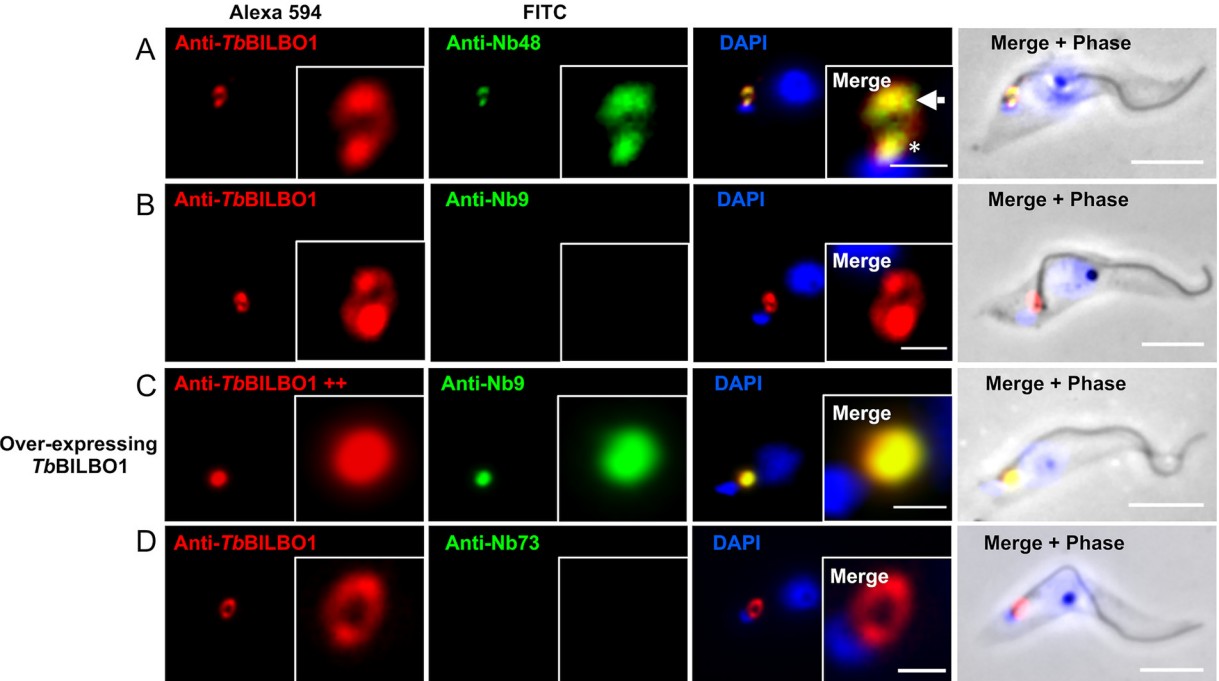

**FIG 1** Nanobodies as TbBILBO1 immunofluorescence detection tools. (A) IFA testing of purified Nb48$_{::HA::6His}$ (green) on wild-type (WT) T. brucei cytoskeletons showing colocalization with TbBILBO1 (probed with anti-TbBILBO1 aa 1-110 rabbit; red). The arrow indicates the FPC and the asterisk indicates colabeling on the MTQ; both of these structures are colabeled. (B) Purified Nb9$_{::HA::6His}$ does not label endogenous TbBILBO1 in WT cytoskeletons. (C) Nb9$_{::HA::6His}$ on T. brucei overexpressing TbBILBO1 (++) shows colocalization with TbBILBO1. (D) Purified Nb73$_{::HA::6His}$ on WT cytoskeletons does not label TbBILBO1. Scale bar is 5$\mu$m, inset is 1$\mu$m. For the sake of convenience, the proteins being probed in all figures are named anti-Nb, plus the relevant nanobody name rather than its tag.

observed on the FPC and the microtubule quartet (MtQ), a subset of specialized microtubules that are in close proximity to the FPC, but with unknown function (27) (Fig. 2A and B). Labeling with anti-TbBILBO1 (rabbit polyclonal, 1-110) alone also clearly labeled the MtQ (Fig. 2C). Colocalization was observed in double labeling experiments with anti-TbBILBO1 (1-110) and Nb48$_{::HA::6His}$ (Fig. 1A and Fig. 2D). In Figures 2A to D, the asterisks indicate the labeling of the MtQ. Purified Nb48$_{::HA::6His}$ and Nb9$_{::HA::6His}$ both positively label full-length TbBILBO1 protein from T. brucei whole-cell extracts in Western blotting experiments (Fig. 2E and F) as well as TbBILBO1 truncated versions (T3$_{::3cMyc}$, aa 171 to 587, and T4$_{::3cMyc}$, aa 251 to 587; please also refer to the schematic in Figure 2G for an overview of the truncations). Neither Nb48$_{::HA::6His}$ nor Nb9$_{::HA::6His}$ labeled T1 or T2, indicating that Nb48$_{::HA::6His}$ and Nb9$_{::HA::6His}$ bind to TbBILBO1 in the region ranging from aa 251 to aa 587. Controls for IFA experiments in Figures 1 and 2 are presented in Figure S2C.

**Surface plasmon resonance confirms a strong binding affinity of Nb48 to TbBILBO1.** To determine the equilibrium dissociation constant ($K_D$) of Nb48$_{::HA::6His}$ (group 1) and Nb9$_{::HA::6His}$ (group 2) to TbBILBO1, we used surface plasmon resonance (SPR) (Fig. 2H and I) (note: it was not possible to assess Nb73$_{::HA::6His}$ [group 3] using SPR due to the inconsistent yield and impure production of this nanobody from bacteria). Sensor chips were coated with purified $_{6His::}$TbBILBO1 by amine coupling and probed sequentially at increasing concentrations with purified Nb9$_{::HA::6His}$ and Nb48$_{::HA::6His}$. The Nb binding results are shown on the sensorgrams (Fig. 2H and I), respectively, and are from a titration data set of three concentrations of the nanobodies: 31.3 nM, 125 nM, and 500 nM. These results illustrate that Nb9$_{::HA::6His}$ and Nb48$_{::HA::6His}$ bind to $_{6His::}$TbBILBO1 but do not behave kinetically in the same way. The superposition of the sensorgrams with the fitting curves validates the Langmuir 1:1 model of interaction for analyzing them. Nb9$_{::HA::6His}$ displays higher rate constants than Nb48$_{::HA::6His}$ with Nb9$_{::HA::6His}$ $k_a$ of $6.54 \times 10^6 \pm 0.27 \times 10^6$ M$^{-1}$ s$^{-1}$ and $k_b$ of $98.9 \times 10^{-3} \pm 0.09 \times 10^{-3}$ s$^{-1}$ and Nb48$_{::HA::6His}$ $k_a$ of $1.74 \times 10^5 \pm 0.16 \times 10^5$ M$^{-1}$ s$^{-1}$ and $k_b$ of $1.52 \times 10^{-3} \pm 0.02 \times 10^{-3}$ s$^{-1}$. Because of a 100-fold lower rate of dissociation and despite a 10-fold lower rate of association, Nb48$_{::HA::6His}$ displays the highest affinity for $_{6His::}$TbBILBO1

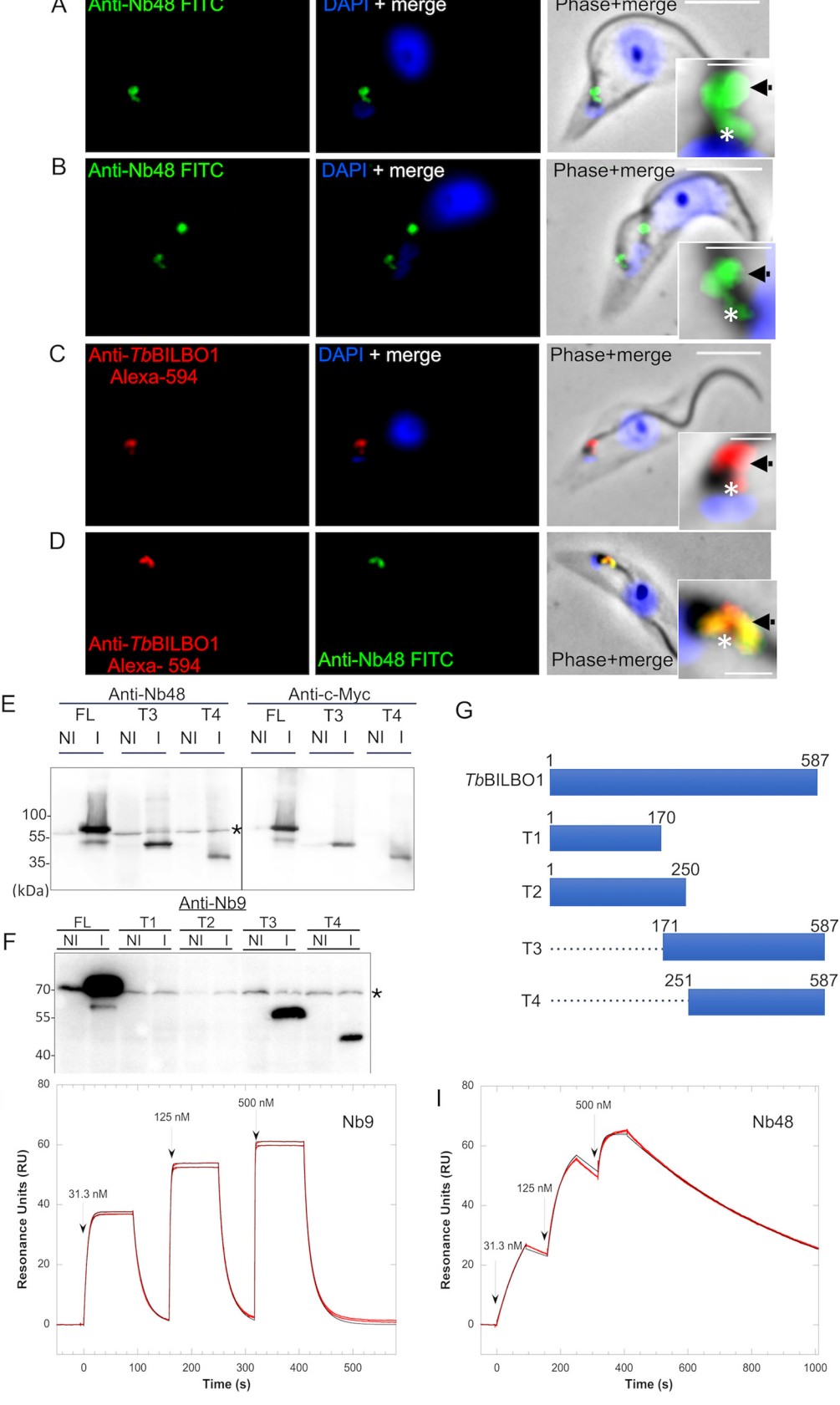

**FIG 2** Anti-*Tb*BILBO1 nanobodies Nb48$_{::HA::6His}$ and Nb9$_{::HA::6His}$ bind to *Tb*BILBO1 *in vitro*. Panels A and B show WT, PCF cytoskeletons probed with Nb48$_{::HA::6His}$ followed by anti-HA, then FITC-conjugated anti-mouse. (A) A parasite cell in the

with a $K_D$ of 8.8 $\pm$ 0.7 nM compared to Nb9$_{::HA::6His}$ $K_D$ of 15.2 $\pm$ 0.7 nM. These values confirm that Nb48$_{::HA::6His6\times His}$ has the highest affinity for $_{6His::}$TbBILBO1. Nb48 provided the best results as a tool by IFA, as a probe on Western blotting, and binding to TbBILBO1 as measured by SPR; we therefore decided to focus on Nb48 for further studies.

**Anti-TbBILBO1 intrabodies bind to the FPC and the MtQ.** Previous studies using stimulated emission depletion microscopy (STED), a technique of switching off the fluorescence of dye molecules by stimulated emission using laser light (49), and immuno-electron microscopy have illustrated that the anti-TbBILBO1 (1-110) rabbit polyclonal antibody labeled both at the FPC and the MtQ. It is important to note here that Esson et al., Albisetti et al., and Dong et al. have shown that TbMORN1, TbSPEF1 (a microtubule binding and bundling protein), and TbSAF1 are localized to the MtQ of flagella, thus clearly indicating that they remain attached to this structure after harsh extraction and indicating that labeling of the MtQ is not artefactual (22, 27, 28). Given that TbBILBO1 and TbMORN1 are stable on isolated flagella, we questioned whether the FPC could be probed *in vivo* in trypanosomes by intrananobody (INb) targeting. Consequently, the DNA sequence encoding Nb48 plus a C-terminal 3cMyc tag was cloned into a *T. brucei* expression vector allowing the tetracycline-inducible expression of the recombinant intrabody 48 (INb48$_{::3cMyc}$). Isolated flagella derived from procyclic cells (PCF) expressing INb48$_{::3cMyc}$ for 12 h were prepared for IFA and double labeled by probing with anti-cMyc and anti-TbBILBO1 (1-110), showing colocalization of TbBILBO1 and INb48 (Fig. 3A). For convenience, the use of INb48 is displayed in these figures as anti-INB48 rather than indicating the tag used on the intrabody. Immuno-electron microscopy was also carried out on flagella that were double labeled using anti-cMyc and anti-TbBILBO1 antibodies (Fig. 3B). Both the IFA and the electron micrographs of these experiments revealed that INb48$_{::3cMyc}$ colocalizes with anti-TbBILBO1 on the FPC but also on the proximal region of the MtQ as described earlier in the IFA studies shown in Figures 2A to D. We also carried out a temporally shorter expression assay of INb48$_{::3cMyc}$ which revealed that it is observed on the FPC and the MtQ at 3 h postinduction (Fig. S2D). Since INb9 also localized with TbBILBO1 on the FPC, it was also tested at reduced expression time (6 h) and was shown on the FPC and colocalized with anti-TbBILBO1 (Fig. S2E).

**The expression of anti-TbBILBO1 intrabodies is cytotoxic in *T. brucei*.** Because anti-TbBILBO1 intrabodies colocalize with TbBILBO1 on isolated flagella, we hypothesized that INbs bind tightly to TbBILBO1 and, as a consequence, may disturb FPC formation or function. To test this hypothesis, we monitored the effect of expression of INb48$_{::3cMyc}$ in PCF *T. brucei* using different concentrations of tetracycline (Fig. 4A). Tetracycline concentrations of 0.1 ng/ml and 1 ng/ml had no effect on cell growth compared to that of the noninduced cells ($-$Tet; red line). However, concentrations from 5 ng/ml to 1 $\mu$g/ml induced a rapid growth defect from 24 h postinduction (hpi) and cell death after 24 hpi. The change in growth rates related to different doses of tetracycline indicates that the lethality of INb48$_{::3cMyc}$ expression in PCF is dose dependent. Considering these results, we chose a tetracycline

**FIG 2** Legend (Continued)

1K1N stage of the cell cycle. (B) A cell in the 2K1N stage. Both cell stages show Nb48$_{::HA::6His}$ labeling on the FPC (black arrow) and MtQ (white asterisk). The MtQ signal was not always apparent on every cell imaged, which could be due to modulations between cell cycle stages. (C) WT, *T. brucei* cytoskeleton probed with anti-TbBILBO1 (1-110) followed by anti-rabbit Alexa fluor 594. (D) WT cytoskeleton probed with anti-TbBILBO1 and Nb48$_{::HA::6His}$ showing complete colocalization. (E) Western blotting of whole-cell extracts of *T. brucei* PCF cells expressing TbBILBO1 full-length (FL) and TbBILBO1 truncations, probed with Nb48$_{::HA::6His}$. Noninduced (NI) and induced (I) trypanosomes expressing TbBILBO1$_{::3cMyc}$ FL protein, aa 171 to 587 (T3), and aa 251 to 587 (T4) were positive when probed with purified Nb48$_{::HA::6His}$. (F) Western blotting of whole-cell extracts of *T. brucei* PCF cells expressing TbBILBO1 full-length (FL) and TbBILBO1 truncations probed with Nb9$_{::HA::6His}$. Noninduced (NI) and induced (I) trypanosomes expressing TbBILBO1$_{::3cMyc}$ FL protein, aa 171 to 587 (T3), and aa 251 to 587 (T4) were positive when probed with purified Nb9$_{::HA::6His}$. Note: all proteins are running slightly faster than predicted size; FL should run at 70 kDa, T3 should run at 49 kDa, and T4 should run at 40 kDa. The black asterisk indicates labeling of endogenous TbBILBO1 seen across all samples. (G) A schematic diagram of cMyc-tagged TbBILBO1 full-length (FL) and truncations. Note that T1 and T2 are soluble. Nb9$_{::HA::6His}$ and Nb48$_{::HA::6His}$ bind to a region within aa 251 and 587 of TbBILBO1. (H) Kinetic analysis by surface plasmon resonance (SPR) of Nb9$_{::HA::6His}$ and (I) Nb48$_{::HA::6His}$, binding to $_{6His::}$TbBILBO1. The results of Nb9$_{::HA::6His}$ and Nb48$_{::HA::6His}$ binding are shown on the sensorgrams, respectively, and are represented by the red curves. The black lines represent the theoretical fit of each Nb as obtained from the Biaevaluation software to a kinetic titration data set of three concentrations of the nanobodies *in vitro*. Nb9$_{::HA::6His}$ displays a good affinity for $_{6His::}$TbBILBO1 with a $K_D$ of 15.2 $\pm$ 0.7 nM; however, Nb48$_{::HA::6His}$ displays the highest affinity for $_{6His::}$TbBILBO1 with a $K_D$ of 8.8 $\pm$ 0.7 nM. Scale bar is 5 $\mu$m, inset is 1 $\mu$m.

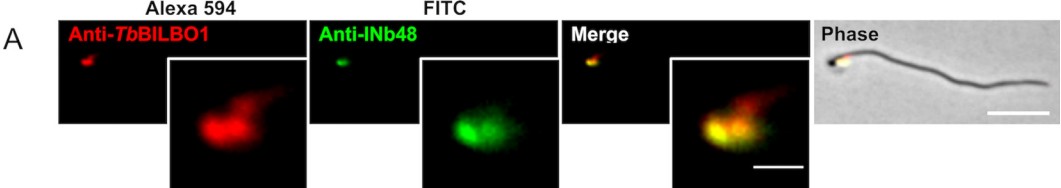

**FIG 3** INb48$_{::3cMyc}$ binds to TbBILBO1 *in vivo*. (A) IFA colocalization of INb48$_{::3cMyc}$ with *Tb*BILBO1 on isolated *T. brucei* PCF flagella. Scale bar of 5 $\mu$m, inset of 1 $\mu$m. (B) Immuno-electron micrographs of a purified flagellum from a trypanosome expressing INb48$_{::3cMyc3}$ for 24 hpi showing clear colocalization of INb48$_{::3cMyc}$ (5 nm gold beads, white arrow) with *Tb*BILBO1 (15 nm gold beads, black arrow). Both 5 nm and 15 nm gold beads are also clearly visible on the MtQ. BB, basal body; PBB, probasal body; FPC, flagellar pocket collar; MtQ, microtubule quartet; A, axoneme. Scale bar is 500 nm, inset is 250 nm.

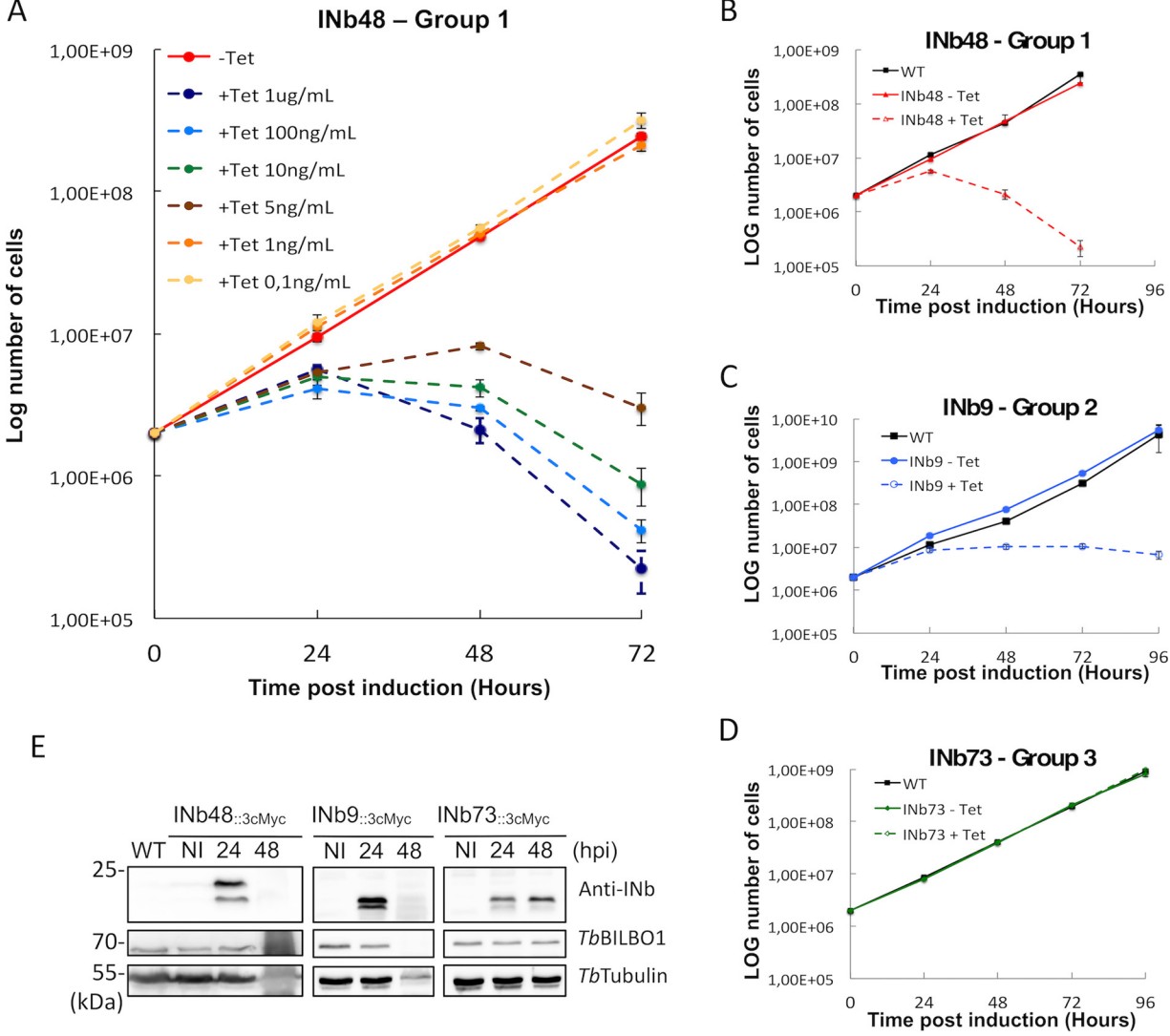

**FIG 4** INb48$_{::3cMyc}$ expression is trypanocidal. (A) Growth curves of trypanosome cells expressing INb48$_{::3cMyc}$. Tetracycline dose-dependent effect on trypanosome growth and cytotoxicity for INb48$_{::3cMyc}$ expression in PCF *T. brucei*. (B) INb48$_{::3cMyc}$ (group 1) expressed in PCF induces rapid decline in population growth followed by cell death, (C) INb9$_{::3cMyc}$ (group 2) expressed in PCF induces cell growth arrest and cell death, and (D) INb73$_{::3cMyc}$ (group 3) expressed in PCF has no effect on population growth. (E) Western blot of INb48$_{::3cMyc}$, INb9$_{::3cMyc}$ and INb73$_{::3cMyc}$ expression in PCF showing expression only in induced cells. Rapid degradation of protein due to cell death at 48 hpi is observed for INb48$_{::3cMyc}$ and INb9$_{::3cMyc}$. Time postinduction is in hours.

concentration of 1 $\mu$g/ml as the standard concentration for INb48$_{::3cMyc}$, INb9$_{::3cMyc}$, and INb73$_{::3cMyc}$ induction. After induction of the expression of these three INbs in trypanosomes, daily samples were taken for cell counts (Fig. 4B to D) and Western blotting (Fig. 4E). Independent induction of each INb resulted in phenotypes that ranged from rapid cell death from 24 hpi for INb48$_{::3cMyc}$ (Fig. 4B) to a reduction in cell growth rate for INb9$_{::3cMyc}$ (Fig. 4C) or no change in growth rate for INb73$_{::3cMyc}$ (Fig. 4D). Western blotting of INb expression is shown in Figure 4E and illustrates robust expression of all INbs at 24 h postinduction in trypanosomes. After 48 hpi of intrabody expression, the signals for INb48$_{::3cMyc}$ and INb9$_{::3cMyc}$ were lost and the signal for *Tb*BILBO1 and tubulin control proteins showed signs of degradation, indicating some degree of cell death in the population. However, no degradation was observed in cells expressing INb73$_{::3cMyc}$, which, importantly, showed no effect on cell growth. As a negative control, an intrabody against green fluorescent protein (INbGFP$_{::3cMyc}$) was expressed in trypanosomes, and as expected, no cell growth defect or phenotype was observed despite its strong expression and cytoplasmic localization (Fig. S3A to C). These data demonstrate that expression of INb48$_{::3cMyc}$ and INb9$_{::3cMyc}$ are

lethal, with INb48$_{::3cMyc}$ showing the strongest effect, while expression of INb73$_{::3cMyc}$ had no effect on cell growth. The data also suggest that INb48$_{::3cMyc}$ and INb9$_{::3cMyc}$ bind to *Tb*BILBO1 *in vivo* and somehow hinder its function.

**Intrabody 48 (INb48$_{::3cMyc}$) binds to *Tb*BILBO1 and induces *Tb*BILBO1 RNAi knockdown phenotypes.** Trypanosome cell cycle stages can be defined based on the arrangement and number of nuclei and mitochondrial genome (kinetoplasts) present in the cell, and wild-type cells should never possess more than the 2K2N DNA complement (50). RNAi knockdown of *Tb*BILBO1 in cultured PCF cells results in cell cycle arrest at the 2K2N stage, producing cells with elongated posterior ends, new flagella that were detached from the length of the cell body, and inhibition of new FP formation (13). Figure 5A shows noninduced (−Tet, NI), whereby the NI trypanosomes display the classic immunofluorescence signal for *Tb*BILBO1 with the annular labeling at the FPC and along the MtQ between the FPC and the basal bodies (BB). After 24 hpi of INb48$_{::3cMyc}$ (+24 h), *Tb*BILBO1 RNAi-like morphological phenotypes were observed where the new flagellum was detached from the length of the cell body and cells demonstrated elongated posterior ends. In *Tb*BILBO1 RNAi knockdown experiments, very little BILBO1 protein was observed at the new flagellum and only low levels were observed at the old flagella. However, when INb48$_{::3cMyc}$ was expressed, both INb48$_{::3cMyc}$ and *Tb*BILBO1 were sometimes observed at the old FPC and also at the base of the detached new flagellum (Fig. 5B, asterisk). More importantly, rather than the classic annular and MtQ signals observed after anti-*Tb*BILBO1 labeling, we observed both *Tb*BILBO1and INb48$_{::3cMyc}$ signals extending along the cell (Fig. 5B, merged inset). This result demonstrates that no new FPC was formed after INb48$_{::3cMyc}$ expression leading to the *Tb*BILBO1 RNAi-like phenotype and, importantly, that INb48$_{::3cMyc}$ binding to *Tb*BILBO1 disturbs FPC formation.

We decided to investigate the organization of the hook complex (HC), a structure intimately associated with the FPC, by immunolabelling of *Tb*MORN1 (28). The HC is an essential structure positioned distal to the FPC, forming a distinctive hook shape, and is considered to be indirectly involved in the regulation of entry of molecules into the flagellar pocket. Figure 5C illustrates the normal location and hook-like shape of anti-*Tb*MORN1 labeling in noninduced cells. However, at 24 hpi of INb48$_{::3cMyc}$ expression, *Tb*MORN1 labeling was elongated distally along the flagellum, and the hook shape was lost (Fig. 5D). Importantly, little or weak *Tb*MORN1 labeling is observed at the base of the new flagellum, suggesting that HC biogenesis is also strongly affected in cells expressing INb48$_{::3cMyc}$. A consistent observation in cells expressing INb48$_{::3cMyc}$ was that the new flagellum was detached from the length of the cell body, i.e., only attached at the base, an abnormal phenotype strongly resembling the *Tb*BILBO1 RNAi knockdown phenotype (13). We further characterized the detached new flagellum phenotype by colabeling with anti-*Tb*PFR2 antibody (51). *Tb*PFR2 is a protein of the paraflagellar rod (PFR) that is present alongside the axoneme of the flagellum of wild-type *T. brucei* (51) as shown in noninduced cells (Fig. 5E). In induced cells (24 hpi), *Tb*PFR2 labeling was observed along the old flagellum and the newly detached new flagellum (Fig. 5F). In this cell, INb48$_{::3cMyc}$ was observed only at the old FPC, indicating that a new flagellum and PFR was formed in the absence of the FPC. To investigate any effect of INb48$_{::3cMyc}$ expression on the basal bodies (BB), an antibody marker anti-*Tb*BLD10 was used, which was raised against *Tb*BLD10, an essential protein in probasal body biogenesis (52). Figure 5G shows a noninduced *T. brucei* cell with the typical labeling of the mature and probasal bodies. At 24 hpi of INb48$_{::3cMyc}$, the labeling of both the pro- and mature basal bodies of the mother and daughter flagella was unaffected (Fig. 5H). We also checked a longer INb48$_{::3cMyc}$ expression (48 hpi) by IFA and demonstrated that cells were severely compromised. Cells showed cell cycle disruption, including the presence of multiple flagella (Fig. S3D and E). Also, the *Tb*BILBO1 labeling revealed extended structures demonstrating that the shape of the polymers formed by *Tb*BILBO1 were strongly affected and were unable to form the typical annular FPC.

The phenotypes induced after INb48$_{::3cMyc}$ expression in trypanosomes were observed in more detail by transmission electron microscopy (TEM). TEM micrographs of thin sections through the FP (*) of a wild-type cell (Fig. 6A) illustrate that the flagellum passes through this structure and that its transition zone is positioned within the FP lumen (black arrow).

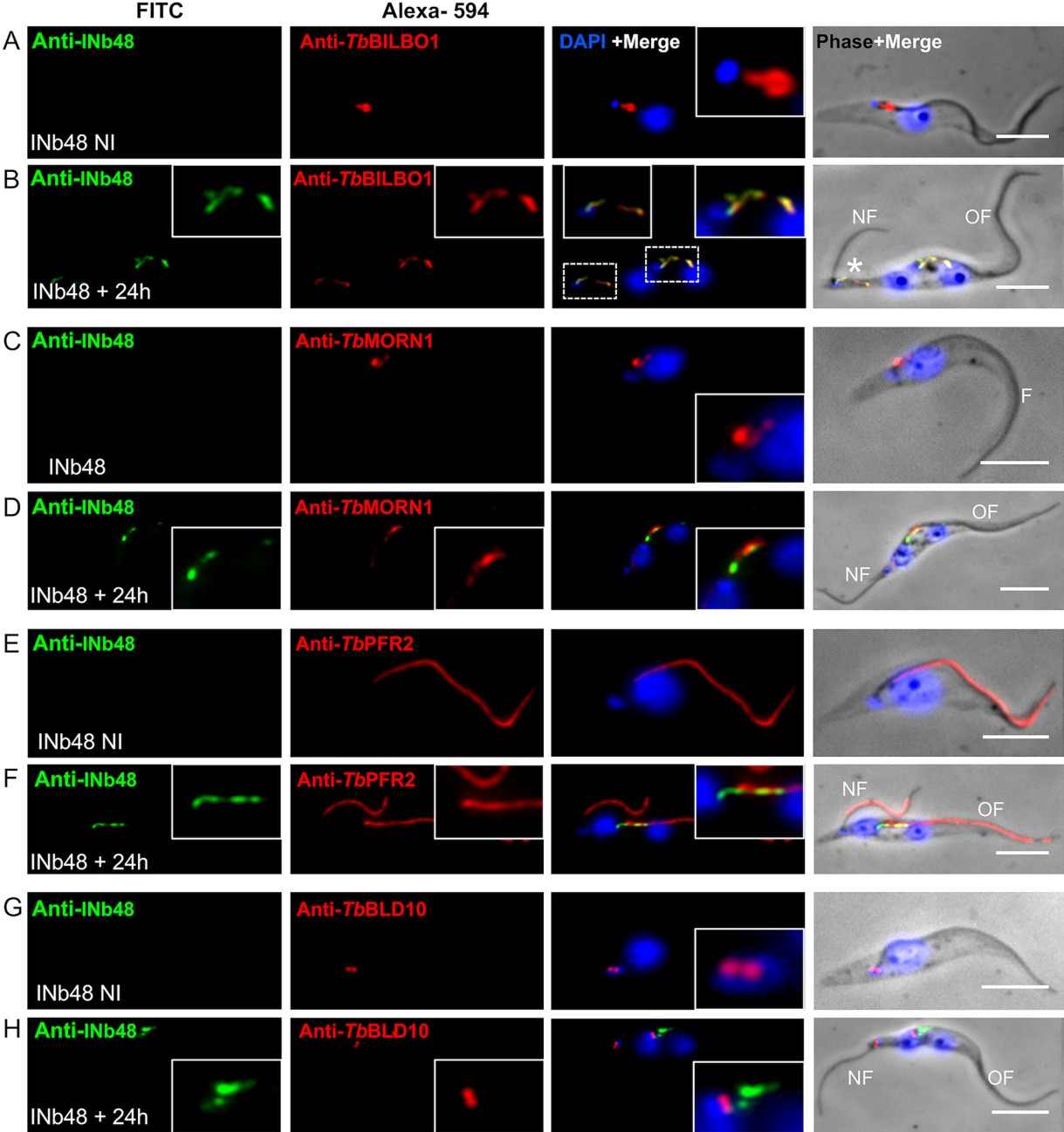

**FIG 5** INb48$_{::3cMyc}$ expression *in vivo* induces phenotypes resembling *Tb*BILBO1 RNAi knockdown. (A) IFA of noninduced (NI) PCF *T. brucei* cells probed with anti-cMyc as a probe for INb48$_{::3cMyc}$ (green) and anti-*Tb*BILBO1 (red), indicating the absence of INb48$_{::3cMyc}$ labeling in noninduced cells. (B) An INb48$_{::3cMyc}$-induced (Ind) *T. brucei* cell at 24 hpi showing colocalization of INb48$_{::3cMyc}$ (green) with a disorganized signal for *Tb*BILBO1 (linear polymer formation instead of the typical annular shape) and a posteriorly localized, detached new flagellum (labeled as NF). Asterisk (*) denotes the presence of colocalization of INb48$_{::3cMyc}$ and anti-*Tb*BILBO1, which is observed on the old flagellum (labeled as OF) and at the base of the new detached flagellum (labeled NF). (C) Noninduced (NI) trypanosome showing correct location of *Tb*MORN1 labeling (red) at the base of the flagellum (labeled as F), with the classic hook and shank composition; there is no anti-cMyc signal in this NI cell. (D) At 24 hpi (Ind), INb48$_{::3cMyc}$ labeling (green) is seen in the location of the FPC with linear labeling extending a short way along the axoneme; *Tb*MORN1 labeling has changed from the classic hook shape and is seen as an extended linear form extending distally from the FPC region along the direction of the old flagellum (OF). (E) Noninduced (NI) trypanosome showing typical labeling of *Tb*PFR2 (red) along the flagellum. (F) 24 hpi Ind INb48$_{::3cMyc}$ (green) showing a trypanosome cell with both the old (OF) and new flagella (NF) labeled with *Tb*PFR2. INb48$_{::3cMyc}$ labeling is observed only at the old FPC extending along anteriorly in this cell. (G) A noninduced (NI) trypanosome cell probed with anti-*Tb*BLD10 (red), labeling both the mature and probasal bodies (BB) at the base of the flagellum. (H) At 24 hpi (+Tet) of INb48$_{::3cMyc}$ (green), a detached new flagellum (NF) is seen at the tip of at an extended posterior end; anti-*Tb*BLD10 labeling is seen associated with both the new and old flagella (red); INb48$_{::3cMyc}$ labeling is seen mainly in the region of the FPC of the old flagellum (green). Scale bar is 5$\mu$m.

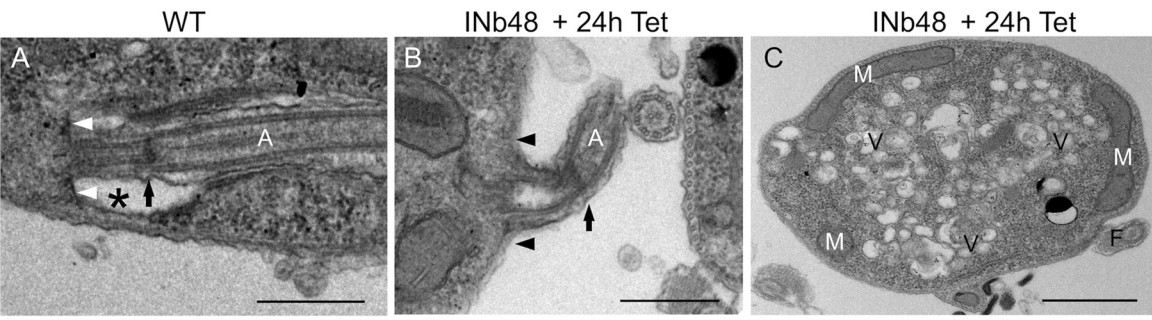

**FIG 6** INb48$_{::3cMyc}$ expression disrupts FPC and HC formation. (A) Transmission electron micrograph (TEM) of a thin section of a WT PCF *T. brucei* cell showing the typical flagellar pocket (FP) (*), the base of the flagellar pocket (white arrows), flagellar axoneme (A), and exit site of the axoneme from the FP membrane; the transition zone is indicated by a black arrow within the FP. (B) Illustrates a detached flagellum from a thin section of PCF cells expressing INb48$_{::3cMyc}$ 24 hpi (+Tet). The FP has not been formed and microtubules are present where the FP membrane should be located (black arrowheads). The transition zone of the axoneme is outside the cell body (black arrow). (C) TEM of a thin section of a cell expressing INb48$_{::3cMyc}$ for 24 hpi (+Tet), showing accumulation of intracellular vesicles. M, mitochondrion; Vm, vesicles; F, flagellum. Scale bar is 1 $\mu$m. (D) Immuno-electron micrograph (iEM) of isolated flagella of PCF *T. brucei* after 24 hpi (+Tet) of INb48$_{::3cMyc}$ expression. INb48$_{::3cMyc}$ is observed (5 nm gold beads represented as black arrowheads) colocalizing with anti-*Tb*BILBO1 (15 nm gold beads represented as black arrows) on the MtQ between the BB and the expected location of the FPC; note that the FPC is absent. (E) iEM of isolated flagella of PCF *T. brucei* after 24 hpi (+Tet) of INb48$_{::3cMyc}$ expression showing INb48$_{::3cMyc}$ (5 nm gold beads represented by black arrowheads) on the MtQ and an abnormal elongated labeling of *Tb*MORN1 (15 nm gold beads represented by black arrows). Note that in trypanosome cells expressing INb48$_{::3cMyc}$, the new FPC is absent and the Hook Complex that is normally represented by *Tb*MORN1 has lost its hook shape. BB, basal body; PBB, probasal body; MtQ, microtubule quartet; A, axoneme; PMtQ, presumed probasal body MtQ. Scale bar is 500 nm, inset scale bar is 100 nm.

INb48$_{::3cMyc}$ expression for 24 h revealed that many of the new flagella were formed in the absence of an FP (Fig. 6B, black arrowheads indicate the position where the FP should be formed), and the transition zone is now external to the cell (black arrows). These cells also accumulated cytoplasmic vesicles (Fig. 6C). These abnormal features are typical of $Tb$BILBO1 RNAi knockdown phenotypes (13) indicating that $Tb$BILBO1 function was indeed hampered by the expression of INb48$_{::3cMyc}$. We isolated flagella from INb48$_{::3cMyc}$ trypanosome cells at 24 hpi (+Tet) and probed them with anti-cMyc (for INb48$_{::3cMyc}$) and anti-$Tb$BILBO1 or, in a separate experiment, anti-cMyc and anti-$Tb$MORN1. The flagella were prepared for negative staining and visualized by transmission electron microscopy (TEM). The colabeling of INb48$_{::3cMyc}$ with $Tb$BILBO1 or $Tb$MORN1 revealed that these latter proteins colocalize on the MtQ (Fig. 6D and E). $Tb$BILBO1 was located on the MtQ proximal to the basal bodies (Fig. 6D), whereas $Tb$MORN1 was primarily on the MtQ distal to the basal bodies (Fig. 6E). Interestingly, however, the typical annular FPC structure was absent in INb48$_{::3cMyc}$-induced trypanosomes, explaining the detached flagellum phenotype. In Figure 6D, INb48$_{::3cMyc}$ (arrowheads) is present alongside $Tb$BILBO1 (arrows) extending from the basal bodies on the MtQ to the location where the FPC should be present. Figure 6E shows $Tb$MORN1 labeling and shows in more detail that in the presence of INb48$_{::3cMyc}$ (arrowheads), it was observed on the MtQ (arrows) extending in a linear fashion proximally along the flagellum but not distally. These data demonstrate that the FPC and HC are structurally perturbed by the expression of INb48$_{::3cMyc}$ in $T. brucei$.

**Expression of INb48$_{::3cMyc}$ disrupts the cell cycle in PCF $T. brucei$.** The organization of kinetoplasts and nuclei in INb48$_{::3cMyc}$-expressing cells was analyzed at different time points (Fig. 7A). The percentage of noninduced (−Tet) cell cycle stages were 74.4% in the 1K1N stage (blue bars), 13.9% in the 2K1N stage, and 9.4% in the 2K2N as published previously (13). A change was observed in trypanosomes expressing INb48$_{::3cMyc}$ for 12 h (+Tet 12 h), with a reduction in 1K1N cells and an increase in 2K1N and 2K2N cells. There was also the appearance of abnormal K/N phenotypes, such as 1K2N (1.5%), when the nucleus had divided before the kinetoplast and cells with other abnormal K/N ratios, including cells with more than 2K and/or 2N, annotated XKXN (1.6%). As mentioned previously, when INb48$_{::3cMyc}$ was expressed in trypanosome cells, they exhibited detached flagella (18.3%). After 24 hpi (+Tet 24 h), this profile continued to augment with a further reduction in 1K1N (44.20%) and a further increase in 2K2N cells (27.5%). Additionally, the number of 1K2N (abnormal) cells increased to 4.3%, and 5.9% of the population showed XKXN phenotype. Finally, the number of cells with detached flagella had also increased to 35.4%. INb48$_{::3cMyc}$-expressing cells in the 2K2N stage were further investigated and categorized according to linear organization of K and N from posterior to anterior orientation (Fig. 7B). For noninduced cells (−Tet), most 2K2N trypanosomes had the normal KNKN organization with both flagella attached along the cell body (green bar). At 12 hpi, the number of normal 2K2N population had decreased (from 93.5 to 45.8%) with the appearance of RNAi $Tb$BILBO1 knockdown-like 2K2N cells. A total of 36.44% of the +Tet 12 h total 2K2N population had detached flagella and an extended posterior end, which also included 15.8% with a K_NKN organization (here the underscore refers to an extended end, orange bar). A total of 13.1% of the abnormal population had a K_KNN kinetoplast organization (red bar), and 7.6% had a poorly segregated nuclei K_NKN* (yellow bar). The other 2K2N population 17.7% (blue bar) included cells that appeared normal in kinetoplast and nuclear organization KNKN but with detached flagella and cells with oddly misplaced kinetoplast (KKNN) with detached flagella. At 24 hpi (+Tet 24 h), the proportion of 2K2N "normal-like" population was significantly reduced to 20.3% at the expense of abnormal cells mentioned above. The rest of the abnormal cells possessed a detached new flagellum which was distributed over a number of arrangements of K and N plus an extended posterior end K_KNN (red bar), K_NKN (orange bar), and K_NKN* (yellow bar) or without extended posterior ends (blue bar). These abnormal phenotypes are a clear indication of disruption of the cell cycle. As with RNAi knockdown of $Tb$BILBO1, it appears that there is a general cell cycle arrest at the 2K2N stage, indicating that the

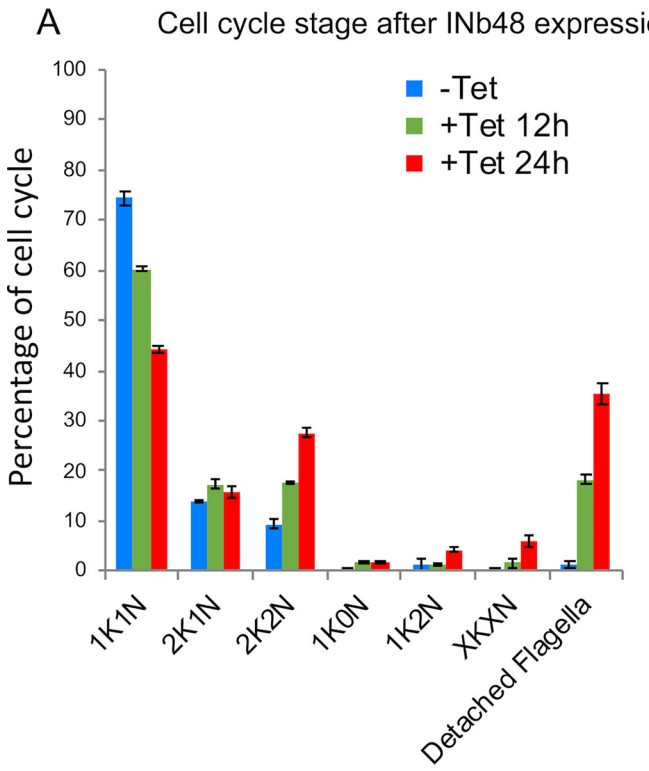

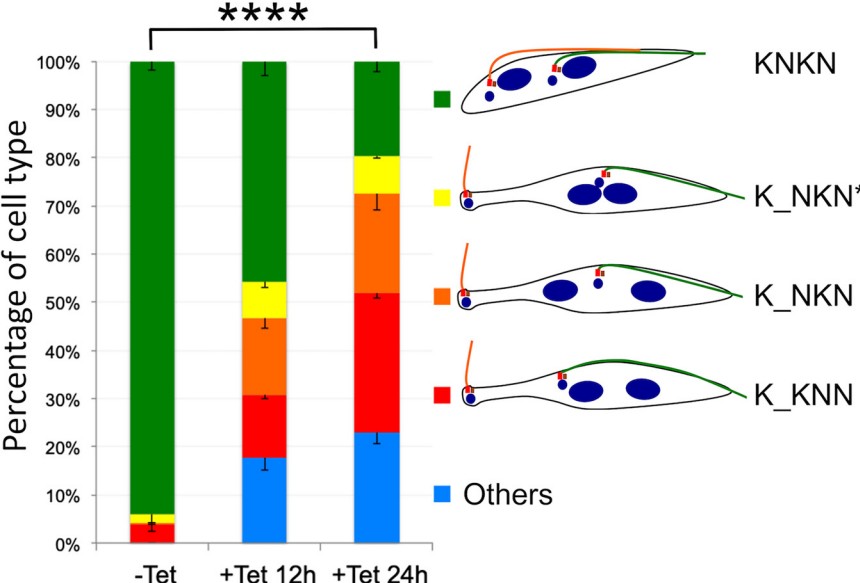

**FIG 7** Expression of INb48$_{::3cMyc}$ disrupts the cell cycle in *T. brucei*. (A) INb48$_{::3cMyc}$ expression disrupts cytokinesis in *T. brucei*. Bar graph to show the percentage of trypanosomes counted at each stage of the cell cycle for noninduced (−Tet, blue bars), 12 hpi (+Tet 12 h, orange bars), and 24 hpi (+Tet 24 h, red bars) for INb48$_{::3cMyc}$ expression. Note that at 24 hpi, approximately 35% of cells have detached flagella. INb48$_{::3cMyc}$ expression leads to a decrease in 1K1N and an increase in 2K1N, 2K2N, abnormal 1K2N, and XKXN phenotypes. *N* is equal to 300 trypanosomes per time point, and each time point has been completed in triplicate. (B) Distribution of 2K2N cells after INb48$_{::3cMyc}$ expression in *T. brucei* showing strong similarities to *Tb*BILBO1 RNAi knockdown. The 2K2N cells were subdivided into 5 categories according to the visual appearance of the cell, i.e., possessing a detached flagellum, an extended posterior end, and abnormal positioning of the kinetoplast and nucleus. ****, $P < 0.0001$.

cells cannot undergo division. With regard to these cell cycle counts, similar findings were observed with RNAi knockdown of *TbBILBO1*, suggesting that the crucial steps and structures required for cytokinesis were impaired after INb48$_{::3cMyc}$ expression and demonstrating that INb48 targeting of *Tb*BILBO1 can be as effective as RNAi knockdown.

**Heterologous expression of *Tb*BILBO1 and INb48.** Expression of full-length and truncated forms of *Tb*BILBO1 in a U-2 OS heterologous system has been published previously (20). That work demonstrated that expression of full-length *Tb*BILBO1 in mammalian cells induces the formation of linear polymers with comma- and globular-shaped termini. To determine if Nb48 can bind to *Tb*BILBO1 as an antibody probe in this heterologous system, we expressed full-length *Tb*BILBO1 in U-2 OS cells and probed detergent-extracted cells with the purified Nb48 (Nb48$_{::HA::6His}$) and with anti-*Tb*BILBO1 (1-110) (Fig. 8B). Nontransfected, extracted U-2 OS cells were probed with anti-TbBILBO1 (1-110) and Nb48$_{::HA::6His}$ and showed a negative signal (Fig. 8A). When full-length *Tb*BILBO1 was expressed, it induced the formation of long BILBO1 polymers that were colabeled with Nb48$_{::HA::6His}$ and anti-*Tb*BILBO1 (1-110) (Fig. 8B). *Tb*BILBO1 possesses two canonical calcium-binding EF-hand domains: 1 and 2. Mutation of EF-hand domain 1 alone, or both EF-hands together, prevented the formation of linear polymers; however, mutation of EF-hand domain 2 alone (Mut2 EF-hand) resulted in the formation of helical polymers (20). We therefore used Mut2 EF-hand to test for colabeling with Nb48. Excellent colabeling was observed when the Mut2 EF-hand was expressed: the helical polymers of the mutated BILBO1 colabel with Nb48$_{::HA::6His}$ (Fig. 8C), suggesting that mutating the calcium-binding site, and preventing calcium binding, does not affect Nb48 binding. We next expressed the *Tb*BILBO1 truncations T1, T2, T3, and T4 (refer to Fig. 2G) in U-2 OS cells and used the previous anti-*Tb*BILBO1 monoclonal IgM antibody to label T3 and T4 constructs. Those previous studies had shown that *Tb*BILBO1 truncations T1 (aa 1 to 170) and T2 (aa 1 to 250) are soluble in U-2 OS cells and do not form polymers (20). No signal was observed when these extracted cells were probed by anti-*Tb*BILBO1 (1-110); furthermore, no Nb48$_{::HA::6His}$ signal was observed (Fig. 8D and E). Truncations T3 (aa 171 to 587) and T4 (aa 251 to 587), however, form long polymers when expressed in U2-OS cells, and as with FL polymers, these were positively colabeled when probed with *Tb*BILBO1 and Nb48$_{::HA::6His}$ (Fig. 8F and G). As such, this suggests that the Nb48 epitope lies between the amino acids 251 to 587. These results illustrate that as a specific nanobody probe, Nb48$_{::HA::6His}$, in a heterologous environment, performed equally to a rabbit polyclonal antibody (anti-*Tb*BILBO1 1-110) and to anti-*Tb*BILBO1 IgM.

To determine if INb48 can bind to *Tb*BILBO1 in the U-2 OS heterologous system and influence polymerization, we coexpressed INb48$_{::3HA}$ with full-length *Tb*BILBO1 (FL), Mut2 EF-hand, or truncated forms of *Tb*BILBO1. Expression of INb48$_{::3HA}$ alone produced a cytoplasmic signal in whole-cell preparations (Fig. 8H) that was eliminated after detergent extraction (Fig. 8I), demonstrating that INb48$_{::3HA}$ does not bind to any cytoskeletal structure in U-2 OS cells and does not form polymers. When INb48$_{::3HA}$ was coexpressed with FL-*Tb*BILBO1 and cells were detergent extracted, a strong colocalization was observed between the *Tb*BILBO1 signal and that of INb48$_{::3HA}$, demonstrating that INb48$_{::3HA}$ was expressed and binds to *Tb*BILBO1, *in vivo*, in extracted conditions (Fig. 8J and K). Importantly, however, the typical polymer structures that *Tb*BILBO1 normally forms in U-2 OS cells (refer to Fig. 8B) were absent. Instead, dense compacted structures attached to thin minor polymers were formed. Similar dense structures were observed when Mut2-EFh was coexpressed with INb48$_{::3HA}$, indicating that the EF-hand domain 2 mutation (which prevents calcium binding) did not influence binding of INb48$_{::3HA}$ to *Tb*BILBO1 (Fig. 8L). These data indicate that binding of INb48$_{::3HA}$ to *Tb*BILBO1 modifies and reduces linear polymerization in this environment.

When INb48$_{::3HA}$ was coexpressed with T1 and with T2 (both have been demonstrated to be cytoplasmic in nonextracted cells) (20), a cytoplasmic signal was observed (Fig. 8M and N). In contrast to the long, linear polymers observed when T3 alone was expressed (Fig. 8F), coexpression of T3 and INb48$_{::3HA}$ induced dense globular structures (Fig. 8O), similar to the observations made when INb48$_{::3HA}$ was coexpressed with FL-*Tb*BILBO1 (Fig. 8K) or Mut2-EFh (Fig. 8L). Coexpression of INb48$_{::3HA}$ with T4 resulted in long polymers similar, but not identical, to expression of T4 alone (as observed in Fig. 8G). Instead, the INb48$_{::3HA}$-bound T4

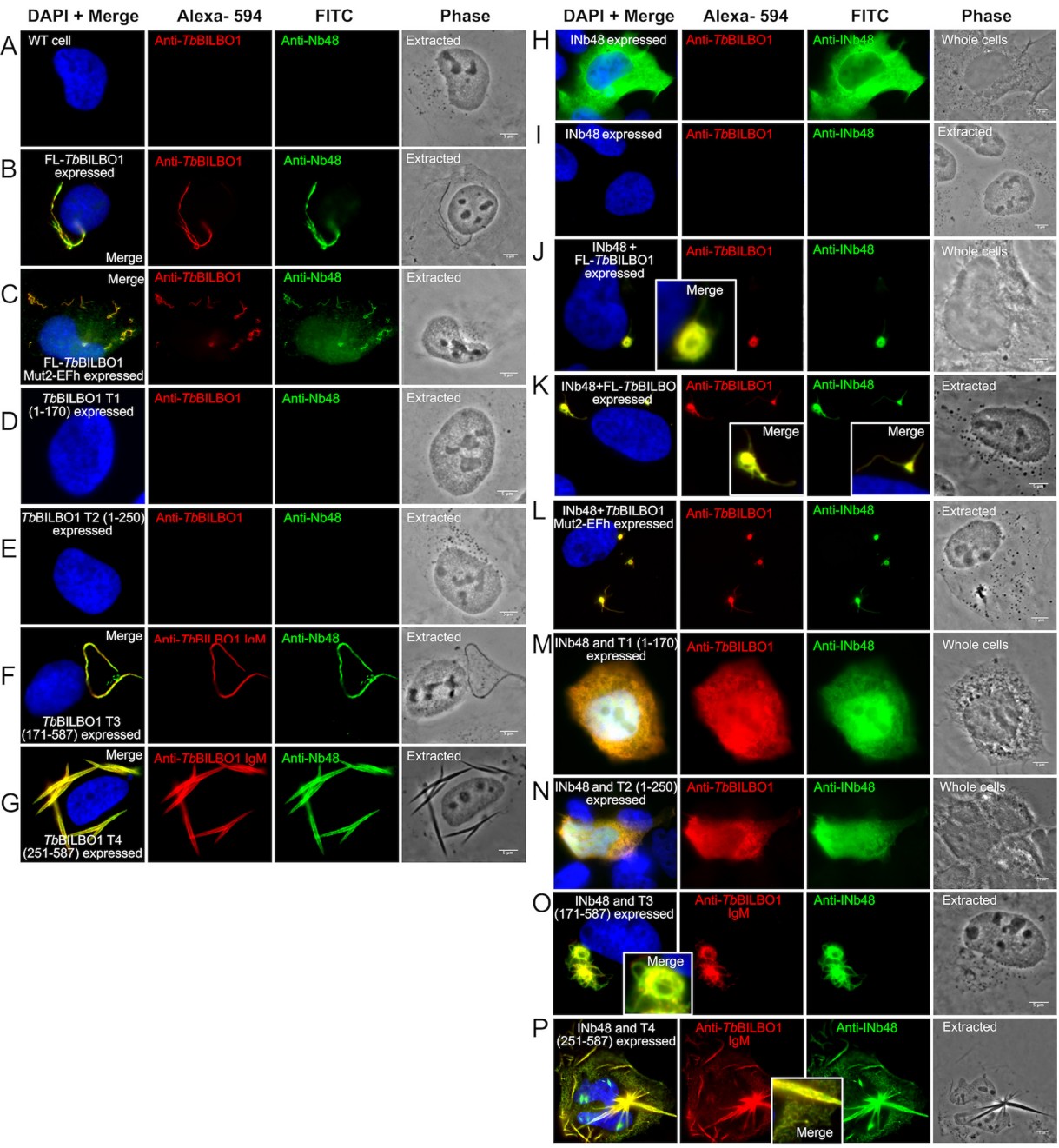

**FIG 8** Nb48 and INb48$_{::3HA}$ bind to *Tb*BILBO1 in a heterologous system, and INb48$_{::3HA}$ interferes with *Tb*BILBO1 polymer formation. (A) A nontransfected U-2 OS cell extracted with Triton X-100, probed with anti-*Tb*BILBO1 (1-110 rabbit) and purified Nb48$_{::HA::6His}$ and revealed independently by anti-HA (mouse) and their respective secondary fluorophores: anti-rabbit, Alexa fluor 594 and anti-mouse, FITC-conjugated. No labeling is observed. (B) A U-2 OS cell that is expressing FL *Tb*BILBO1 and probed as described above. *Tb*BILBO1 forms polymers and Nb48$_{::HA::6His}$ labels these polymers. (C) A U-2 OS cell expressing mutated EF-hand domain 2 (Mut2EF-*Tb*BILBO1) and probed as described above. The typical spiral polymers of this mutant are formed and are colabeled with rabbit 1-110 and Nb48$_{::HA::6His}$. (D and E) *Tb*BILBO1 truncations T1 (aa 1 to 170) and T2 (aa 1 to 250), which are both cytoplasmic and thus eliminated during detergent extraction and thus not labeled. (F and G) The typical signals observed when *Tb*BILBO1 truncations T3 (171 to 587 aa) and T4 (251 to 587 aa) are expressed in U-2 OS; the resulting polymers formed are both positive for *Tb*BILBO1 (in this case anti-*Tb*BILBO1 mouse monoclonal, IgM) and Nb48$_{::HA::6His}$ (anti-HA rabbit, IgG). The IgM was used here because it binds to the C terminus whereas anti-*Tb*BIL1BO (1-110) binds to the N terminus of *Tb*BILBO1. (H to P) In all experiments, INb48, *Tb*BILBO1, and *Tb*BILBO1 mutations and truncations were expressed in U-2 OS cells for 24 h and then processed for IFA. IFA of nonextracted U-2 OS cell expressing INb48$_{::3HA}$ only, and probed with anti-*Tb*BILBO1 and anti-HA to detect the intrabody. This cell shows that INb48$_{::3HA}$ is cytoplasmic. (I) Illustration of a cell expressing INb48$_{::3HA}$ only and probed with anti-*Tb*BILBO1 and anti-HA after detergent extraction, showing that INb48$_{::3HA}$ has been removed after extraction. (J) Illustration of a nonextracted cell expressing INb48$_{::3HA}$ and FL-*Tb*BILBO1 simultaneously and probed as in panels H and I. It illustrates that INb48$_{::3HA}$ binds to the *Tb*BILBO1 polymers. However, these polymers are different from WT polymers shown in panel B, suggesting that INb48$_{::3HA}$ affects normal *Tb*BILBO1 polymer formation. (K) A cell treated as in panel J but detergent extracted, demonstrating that INb48$_{::3HA}$ is not removed upon detergent extraction.

polymers were associated with many small, detergent-insoluble, network-forming polymers (Fig. 8P), which were never observed in T4-only expression as described previously (20) and suggest that binding of INb48::3HA to T4 modifies normal polymerization. Taken together, these data suggest a critical disturbance by INb48 in *Tb*BILBO1 polymer organization when coexpressed in a heterologous mammalian system. IFA controls for INb48::3HA and *Tb*BILBO1 expression in U-2 OS cells alone and together can be found in Figure S4A to D, demonstrating a lack of labeling when no primary antibodies were used. Figure S5A to E shows the control intrananobody against green fluorescent protein (INbGFP) expressed in U-2 OS cells exhibiting a cytoplasmic signal which was extracted in cytoskeleton preparations and did not colocalize or bind to *Tb*BILBO1 polymers or any other structure. Negative and no primary antibody controls are also shown.

## DISCUSSION

Nanobodies are rapidly being developed, not only as revolutionary therapeutics but also as biological tools. In mammalian cells, they have been used to assess protein function intracellularly (intrabodies), to explore protein-protein interactions, and as protein inhibitors (42, 53–56). Nanobodies have been used in a number of experiments associated with trypanosomes, including the potential of nanobodies as diagnostic tools, in experimental therapy, and as trypanolytic agents (42, 57–59). However, intrabodies have not been used to target the trypanosome cytoskeleton intracellularly as a knockdown tool. This study shows that Nb48 proved to be an excellent tool for detecting its trypanosomatid target antigen *Tb*BILBO1 by Western blotting and by immunofluorescence (IFA) on fixed trypanosomes and immuno-electron microscopy. Further, when expressed as an intrabody, INb48 targeted specifically *Tb*BILBO1 and functioned to disrupt the role of this essential cytoskeletal protein.

The difference in the IFA behavior between Nb48 and Nb9 may be explained by their CDR3 sequences. If we consider that the two target epitopes are clearly different, then this can, in part, explain a difference in detection by IFA. This would also suggest that binding of Nb48 to its BILBO1 epitope is possible under the *in vivo* native BILBO1 expression levels within the FPC, whereas binding of Nb9 is impaired due to the possible presence of BILBO1 partner proteins that could partially shield the epitope or because the epitope is not fully exposed. Overexpression of BILBO1 would most probably produce and expose much more unshielded and/or accessible BILBO1 epitope allowing Nb9 to bind. This would also explain why Nb9 was initially selected using enzyme-linked immunosorbent assay (ELISAs) since the selection protocol was based on the use of purified, denatured recombinant BILBO1.

**INb48 modifies *Tb*BILBO1 polymers in U-2 OS cells and in *T. brucei*.** Nb48 and INb9 both bind to the T4 domain of *Tb*BILBO1 (aa 251 to 587), demonstrating that they recognize the same epitope *in vitro* and *in vivo*. Binding of INb48 to *Tb*BILBO1 in U-2 OS cells modifies polymer structure and induced condensed structures (Fig. 8J and K) compared to the long, comma-shaped, or annular-shaped polymers that are observed normally (20). INb48 binds to the domain of *Tb*BILBO1 that includes the coiled-coil domain (CCD) and the leucine zipper (LZ) domain both involved in protein-protein interaction, and importantly,

**FIG 8 Legend (Continued)**

(L) The cell in this image is simultaneously expressing an EF-hand domain 2 mutated form of *Tb*BILBO1 (Mut2-EFh) and INb48::3HA. The typical helical spirals formed by the Mut2-EF-hand protein are not formed when bound by INb48::3HA, suggesting again that INb48::3HA binding affects polymer formation. *Tb*BILBO1 proteins and INb48::3HA were expressed in U-2 OS cells for 24 h; in this case, INb48::3HA was coexpressed with T1 to T4 truncations. Cells were then processed for IFA using either the anti-*Tb*BILBO1 (1-110) rabbit polyclonal, raised to and binding only the N terminus of *Tb*BILBO1, for detecting T1 and T2, or an in-house-made anti-*Tb*BILBO1 IgM mouse monoclonal, which recognizes the complement terminus of *Tb*BILBO1; anti-HA was used to detect INb48::3HA followed by their respective secondary antibodies. (M) An IFA image of a U-2 OS cell simultaneously expressing INb48::3HA and *Tb*BILBO1 T1. These images illustrate that INb48::3HA and T1 are both cytoplasmic and therefore give an overlapping signal. (N) An IFA image of a U-2 OS cell simultaneously expressing INb48::3HA and *Tb*BILBO1 T2. Similar to panel M, both of these proteins are cytoplasmic and give an overlapping signal. (O) This is an IFA image of a U-2 OS cell simultaneously expressing INb48::3HA and *Tb*BILBO1 T3. The typical linear filaments formed by T3 are modified when bound by INb48::3HA, suggesting that the intrabody binding inhibits the typical polymer formation. (P) Detergent-extracted IFA images of U-2 OS cells simultaneously expressing INb48::3HA and *Tb*BILBO1 T4. The spindle shaped polymers, as observed in Figure 8G, are observed to a lesser extent, but numerous smaller polymers are present in the cytoplasm. These are never present in cells expressing T4 alone, suggesting that the binding of INb48 to T4 modifies the polymer-forming capacity of this truncation. Scale bar is 5 $\mu$m.

the CCD is required for dimerization and the LZ domain is required for polymerization (20, 24). Based on the structures observed in the U-2 OS cells, both dimerization and polymerization occur, but it is difficult to speculate on how INb48 affects the polymers. However, one could expect the same effect in parasites, resulting in alteration in FPC formation. Indeed, the expression of INb48 in *T. brucei* led to perturbation of the structure of the FPC and the HC, as seen by IFA and immuno-electron micrograph (iEM) labeling of *Tb*BILBO1 and *Tb*MORN1, respectively. Interestingly, INb48 labeled strongly the MtQ, but the typical annular FPC structure and "hook and shank" aspect of the HC were not observed in many salt-extracted flagella in iEM experiments. Also, IFA experiments demonstrated (via the elongated structures labeled with an anti-BILBO1 polyclonal) that INb48 affects the biogenesis of the FPC. This suggests that *Tb*BILBO1 and *Tb*MORN1 traffic along the MtQ.

**INb48 induces BILBO1-RNAi knockdown-like phenotypes.** The perturbations of *Tb*BILBO1 assembly by INb48 led to RNAi knockdown-like phenotypes, and this included similar disruption to the endomembrane system. Although membrane trafficking was affected by INb48 expression, it does not have membrane targeting or retention signals, which are sometimes used on INbs. This suggests that the accumulation of intracellular vesicles following INb48 expression (Fig. 6) is due to disruption in endo-/exocytosis and is a downstream consequence of the disturbance of the flagellar pocket and the HC. After 24 h of INb48 induction, there was no INb48 or BILBO1 protein degradation and the BILBO1 RNAi-like phenotypes were apparent, but at 48-h induction there was considerable protein degradation probably due to cell lysis. INb48 binding to BILBO1 might result in BILBO1 sequestration, which could lead to intense disruption of the FP and at some point induce cell lysis. It is also possible that the intrabody binds to BILBO1 that has already been incorporated into the FPC but does not disrupt its function. If the intrabody is only able to capture free BILBO1, the difference in effect on free versus FPC-incorporated BILBO1 could explain the delay in the phenotype. In this context, it is noteworthy that we also observed dramatic changes in BILBO1 polymers when they were coexpressed with INb48 in U2-OS cells. The binding of INb48 to newly forming BILBO1 polymers could prevent correct FPC formation by inducing malformation of the growing BILBO1 polymers. Indeed, Figures 5B and D (and Fig. S3D and E) illustrate the formation of unusual, BILBO1-positive, linear polymers when INb48 was expressed in trypanosomes instead of the annular polymers normally present in a wild-type FPC. We are currently investigating the knockdown effect in more detail and also the trafficking of FPC proteins to the FPC to ascertain whether the MtQ is implicated. This may also answer, in part, the question of how the FPC, a flagellum-associated complex of proteins, is built.

In conclusion, we induced a BILBO1 RNAi phenotype not by knocking down protein expression but rather by preventing protein function. This indicates that an intrabody-induced, protein binding approach can prevent the formation of a new FPC most likely even in the presence of BILBO1 interacting proteins. This would indeed suggest that BILBO1 is the main protein performer in FPC biogenesis. Therefore, our data support the hypothesis that targeting minor, yet essential, cytoskeletal proteins is of considerable merit in the search to understand parasite biology. Intrabodies would be useful in organisms that do not have RNAi machinery and to characterize the biology of emerging pathogens, newly discovered protists, and new model organisms (48). As a consequence, the data presented here demonstrate, for the first time, the use of a functional, tractable, anti-cytoskeletal intrabody in *T. brucei* that is able to precisely target and bind to its target epitope and induce disruption of a cytoskeletal structure leading to rapid cell death.

## MATERIALS AND METHODS

**Ethics statement.** The alpaca immunization was carried out by Nanobody Service Facility, VIB, Belgium, and was approved by the Ethical Committee for Animal Experiments of the Vrije Universiteit Brussel (VUB), Brussels, Belgium.

**Alpaca immunization and Nb library construction.** HIS-tagged, full-length *T. brucei brucei* BILBO1 protein ($_{6His::}$*Tb*BILBO1) was expressed in bacteria, purified in urea on nickel nitrilotriacetic acid (Ni-NTA) resin by affinity column purification, as described in reference 25, and sent to Nanobody Service Facility (NSF, VIB, Brussels). The nanobody library was constructed as described by NSF-VIB (Nanobody Service Facility, Belgium) and produced $2 \times 10^8$ independent transformants that were subjected to two rounds of panning, performed on solid-phase coated full-length *Tb*BILBO1 antigen (10 $\mu$g/well). Ninety-five colonies from round two were randomly selected and analyzed by ELISA for the presence of antigen-specific Nbs in their periplasmic

extracts. Thirty-four colonies scored positive, representing seven different Nbs from three different groups based on their gene sequences: Nbs in each group have very similar sequence data. The Nb genes were cloned into a pMECS vector containing a secretion PelB leader signal sequence at the N terminus and a hemagglutinin A (HA) tag and a hexa-histidine (His) tag at the C terminus for purification and immunodetection. These vectors were transformed into TG1 *Escherichia coli* and stored at −80°C.

**Nanobody cloning and bacterial expression.** Nbs were subcloned from pMECS to pHEN6c vector as follows: TG1 *E. coli* harboring the Nbs were grown overnight on solid agar plus ampicillin (100 $\mu$g/ml). The Nb genes were amplified by PCR directly on colonies using specific primers A6E (5′ GAT GTG CAG CTG CAG GAG TCT GGR GGA GG 3′) and PMCF (5′ CTA GTG CGG CCG CTG AGG AGA CGG TGA CCT GGG T 3′). PCR fragments were digested with PstI and *Bst*II and then ligated into the pHEN6c vector harbored in XL1 blue *E. coli* and stored at −80°C. For expression and purification, WK6 *E. coli* cells were transformed with pHEN6c harboring the Nb. PCR on transformed colonies, using M13R (5′ TCA CAC AGG AAA CAG CTA TGA C 3′) and PMCF (5′ CTA GTG CGG CCG CTG AGG AGA CGG TGA CCT GGG T 3′), confirmed the Nb insert. *Tb*BILBO1, *Tb*BILBO1 EF-hand domain mutant 2, and *Tb*BILBO1 truncations T1, T2, T3, and T4 were expressed in the heterologous mammalian system, U2-OS, by using pCDNA3 vector as described in reference 23. Nb48 sequence was cloned into the pcDNA3.1 C terminus tag 3HA vector (derived from pCDNA3.1 Invitrogen) between NheI and XhoI restriction sites and expressed transiently as an intrabody, INb48$_{::3HA}$.

**Intrananobody expression in Trypanosoma.** The Nb inserts were digested out from pHEN6c with HindIII and NdeI and cloned into HindIII and NdeI digested pLew100-3cMyc (60). Plasmid sequences were confirmed by DNA sequencing.

**Nanobody expression and purification.** Freshly transformed WK6 with pHEN6c-Nb vectors was inoculated into 10 ml LB plus ampicillin (100 $\mu$g/ml) plus 1% glucose and incubated overnight at 37°C, shaking at 250 rpm. Terrific broth (TB) medium was prepared for Nb expression (2.3 g/liter KH$_2$ PO$_4$, 16.4 g/liter K$_2$ HPO$_4$.3H$_2$O, 12 g/liter tryptone, 24 g/liter yeast extract, 4% glycerol). One milliliter of culture was added to 330 ml TB plus 100 $\mu$g/ml ampicillin, 2 mM MgCl$_2$, and 0.1% glucose, incubated at 37°C, shaking at 250 rpm, until a mid-log phase of growth optical density at 600 nm (OD$_{600}$) of 0.6 to 0.9 was reached. Nb expression was induced with 1 mM isopropyl $\beta$-D-1-thiogalacto-pyranoside (IPTG). The culture was incubated for 17 h at 28°C with shaking. Nb was extracted from the periplasm as follows. Cultures were centrifuged for 16 min at 4,500 × *g*. The pellet was resuspended in 4 ml of TES (0.2 M Tris [pH 8.0], 0.5 mM EDTA, 0.5 M sucrose) and then shaken for 1 h on ice at 110 rpm. Four milliliters of TES/4 (TES diluted 1:4 in water) was added and further incubated on ice for 1 h with shaking and then centrifuged for 60 min at 4,500 × *g* at 4°C to release the contents of the periplasmic space (61). The supernatant contains the periplasmic extract with Nb. A control periplasmic extraction was performed using the same procedure as above, without the Nb insert in the vector.

The periplasmic extract was filtered on 0.45 $\mu$m and loaded onto a HisTrap FF column (GE Healthcare) pre-equilibrated with running buffer (phosphate-buffered saline [PBS], 20 mM imidazole, 10% glycerol, protease inhibitor cocktail set III, Calbiochem, 539134-1 at 1:10,000 dilution). The column was washed with five column volumes (CV) of lysis running buffer, and bound protein was eluted by 500 mM imidazole/running buffer with 10 CV. Elutions containing the highest concentration of purified nanobody were pooled and dialyzed against 1× PBS plus 20% glycerol using Slide-A-Lyzer (Thermo Scientific B2162132) and stored in 1× PBS plus 20% glycerol. To verify the presence of Nb in purified elutions, 10 $\mu$l from each elution was run on 15% SDS-PAGE gel stained with Coomassie brilliant blue (Instant Blue Expedeon Ltd.). Proteins from a second identically loaded gel were transferred to polyvinylidene difluoride (PVDF) membrane using a semidry transfer method (BIORAD), and Western blotting was performed using anti-His (Sigma H-1029, mouse, 1:3,000) and anti-HA (Biolegend MMS-101R or Santa-Cruz 7392, 1:1,000) antibodies to detect the Nb, followed by anti-mouse horseradish peroxidase (HRP) conjugated (Jackson, 115-035-044, 1:10,000 or Jackson, 515-035-062, 1:1,000).

**Surface plasmon resonance assays.** Surface plasmon resonance (SPR) was used to measure the binding affinity of the nanobodies to *Tb*BILBO1 purified protein, the benefits being that it is measured in real-time and is label free. All solutions and buffers used were filtered and degassed. The experiments were performed at 25°C using a Biacore T200 instrument (GE Healthcare Life Sciences, Uppsala, Sweden) flowed with EP+ buffer (10 mM HEPES [pH 7.4], 150 mM NaCl, 3 mM EDTA, 0.05% Tween 20) from the manufacturer as running buffer. For *Tb*BILBO1 expression, $_{6His::}$*Tb*BILBO1 protein was overexpressed in bacteria and soluble supernatant was purified on Ni-NTA resin as described in reference 13. Briefly, the cells were grown at 37°C in Luria-Bertani (LB) medium containing 50 $\mu$g/ml kanamycin to an OD$_{600}$ of 0.6. The cells were cold-shocked on ice for 30 min and induced with 1 mM isopropyl-$\beta$-D-thiogalactopyranosid (IPTG) for 2 h at 16°C. After centrifugation, the cell pellet was lysed by sonication in lysis buffer (2 mM Tris–HCl [pH 7.4], 0.5 M NaCl, 20 mM imidazole, 10% glycerol, protease inhibitor cocktail set III, Calbiochem, 1:10,000) and centrifuged (10,000 × *g*, 30 min, 4°C) to remove cell debris. The supernatant was filtered at 0.45 $\mu$m and loaded onto a HisTrapTMFF column (GE Healthcare) preequilibrated with the same lysis buffer. The column was washed with 5 column volumes (CV) of lysis buffer, and bound protein was eluted by 500 mM imidazole with 10 CV in lysis buffer. *Tb*BILBO1 fractions were pooled and dialyzed against PBS and 10% glycerol, and concentration was measured using Thermo Scientific NanoDrop 2000. *Tb*BILBO1, prepared at 50 $\mu$g/ml in 10 mM sodium acetate buffer (pH 7) and was immobilized on one flow cell of a CM5 sensor chip (GE Healthcare, lot no. 10266084) by amine coupling as indicated by the manufacturer, and the protein solution was injected for 11 min; 2,973 resonance units (RU) of protein were immobilized. One flow cell was left blank for double-referencing of the sensorgrams. The nanobodies were dialyzed against the EP+ running buffer and were injected over the target by single-cycle kinetics (SCK). This method consists of injecting the partner sequentially at increasing concentrations without a regeneration step between each concentration injected. Protein concentrations were determined using a Thermofisher Nanodrop One spectrophotometer (Ozyme) and were injected over the target by single-cycle

kinetics (SCK) (62, 63). The surface was regenerated after each SCK cycle with two 1-min pulses of 10 mM glycine-HCl (pH 2.1). The sensorgrams were analyzed with Biacore T200-v2.0 evaluation software using a Langmuir 1:1 model of interaction to determine the association ($k_a$) and dissociation ($k_b$) rate constants. A regeneration test was performed using 10 mM glycine-HCl (pH 2.1; GE health care), 10 mM glycine-HCl (pH 2.1), at 30 $\mu$g/min for 30 s with $Tb$BILBO1 alone to ensure no loss of signal. The dissociation equilibrium constant, $K_D$, was calculated as $k_b/k_a$. The values shown are the average and standard deviation of three independent experiments with samples injected in duplicate.

**Trypanosome cell lines, culture, and transfection.** Procyclic (PCF) cell line Tb427 29.13 *Trypanosoma brucei brucei* was grown in SDM-79 medium (pH 7.4; GE Healthcare, G3344-3005) supplemented with 2 mg/ml hemin (Sigma-Aldrich, H-5533), 10% fetal bovine serum (FBS) complement deactivated at 56°C for 30 min (Gibco, 11573397), and incubated at 27°C. Selection antibiotics hygromycin (25 $\mu$g/ml) and neomycin (10 $\mu$g/ml) were added to the media to maintain the plasmid vectors pLew29 and pLew13 (60). For transfection, the pLew100_Nb_3cMyc plasmids were linearized with NotI and then concentrated and purified by ethanol precipitation. Laboratory cell lines PCF Tb427 29.13 were grown to 5 $\times$ 10$^6$ to 10 $\times$ 10$^6$ cells/ml, and then 3 $\times$ 10$^6$ cells were electro-transfected with 10 $\mu$g of plasmid using the program X-001 of the NucleofectorII, AMAXA apparatus (Biosystems) as described in reference 60 with the transfection buffer described therein. After transfection, clones were obtained by serial dilution and maintained in logarithmic phase growth at 2 $\times$ 10$^6$ cells/ml. Selection antibiotics hygromycin (25 $\mu$g/ml) and neomycin (10 $\mu$g/ml) were added to the media to maintain the laboratory cell lines Tb427 29.13 (procyclic forms [PCF]). Phleomycin (5 $\mu$g/ml) was added to the media of transfected cells to select for those harboring the pLew100-3cMyc vector and the Nb gene. Expression of Nb in the parasites (intrabody) was induced with tetracycline at 0.1 ng/ml to 1 $\mu$g/ml tetracycline. Growth curves representing logarithmic number of cells were calculated by counting the number of cells every 24 h using Malassez counting chamber slides. Error bars represent standard error of the mean (SEM).

**Mammalian cell line culture and transfection.** U-2 OS cells (human bone osteosarcoma epithelial cells, ATCC no. HTB-96) (64) were grown in Dulbecco's modified Eagle medium (DMEM) Glutamax (Gibco) supplemented with final concentrations of 10% fetal calf serum (Invitrogen), 100 units/ml of penicillin (Invitrogen), and 0.100 mg/liter of streptomycin (Invitrogen) at 37°C plus 5% $CO_2$. Exponentially growing U-2 OS cells in 24-well plate with glass coverslips were lipotransfected as described in reference 20 with 0.5 $\mu$g of DNA using lipofectamine 2000 in OPTIMEM (Invitrogen) according to the manufacturer's instructions and processed for immunofluorescence 24 h posttransfection.

**Western blotting.** Trypanosomes were collected, centrifuged at 800 $\times$ $g$ for 10 min, and washed in 1$\times$ PBS. For whole-cell samples, the number of trypanosomes to be loaded per well was calculated and then resuspended in protein sample buffer 2$\times$ plus nuclease 250 IU/ml (Sigma-Aldrich, E1014). The samples were boiled at 100°C for 5 min and then stored at $-20$°C until required. We prepared 15% SDS-PAGE gels and loaded samples with 5 $\times$ 10$^6$ WT PCF trypanosome cells or PCF expressing pLew100-Nb-3cMyc (induced with tetracycline 1 $\mu$g/ml, 24 to 48 h). For detection of Nb in bacterial periplasmic extract, 2 $\times$ 10$^8$ bacterial cells or equivalent volume of periplasmic extract, flowthrough, or elution were loaded. Samples were separated according to the manufacturer's recommendations and then transferred in a semidry system (BIORAD trans-blot semidry transfer cell 221BR54560) onto PVDF membrane and incubated with blocking solution (BS; 5% skimmed milk powder, 0.2% Tween 20 in 1$\times$ Tris-buffered saline [TBS]) for 60 min. For detection of intrabody, INb::$_{3cMyc'}$ expression in trypanosomes, an anti-cMyc antibody (Sigma, rabbit, C-3956) was diluted in blocking solution 1:1,000 and incubated with the membranes overnight at 4°C. Membranes were washed in 1$\times$ TBS and then in BS before probing with anti-rabbit antibody conjugated to horseradish peroxidase (HRP) (1:10,000 in BS, Sigma, goat, A9169) for 60 min at room temperature (RT). The following antibodies were also used as primaries: anti-$Tb$BILBO1 aa 1-110 (rabbit polyclonal, 1:1,000) (23). Anti-$Tb$Enolase (rabbit polyclonal, 1:25,000) (65) and anti-TAT1 (anti-tubulin mouse monoclonal, 1:1,000) (66) were used as loading controls. A secondary anti-mouse HRP antibody was also used, diluted 1:10,000 (Jackson, sheep, 515-035-062). Revelation was carried out using Image Quant LAS 4000 (GE) or Chemidoc (BioRad). Purified Nbs were used as primary antibodies in Western blotting to detect the presence of $Tb$BILBO1 in whole trypanosome cells. Whole-cell extract of 5 $\times$ 10$^6$ WT PCF trypanosome cells, or PCF expressing $Tb$BILBO1-3cMyc, full-length or truncations of $Tb$BILBO1 (T1, T2, T3, and T4), induced with tetracycline 1 $\mu$g/ml, 24 h), were loaded on 12 or 15% SDS-PAGE. Western blotting was carried out as described above, and the purified Nb$_{3HA:6His}$ was used at a concentration of 10 $\mu$g/ml to probe the membranes. To detect purified nanobody, an anti-HA (1:1,000, Biolegend, 901513) or anti-His (1:10,000, Sigma H1029) was used, followed by an anti-mouse HRP conjugated secondary (see above).

**Immunofluorescence assay.** One milliliter of log-phase trypanosomes was collected, centrifuged for 5 min at 1,000 $\times$ $g$, and washed in 1$\times$ PBS, and then 20 $\mu$l was loaded onto poly-L-lysine 0.1% solution (P8920 Sigma-Aldrich) coated slides for 4 min to adhere. Whole cells were fixed in methanol at $-20$°C for at least 60 min. Cytoskeleton extraction was with 0.25% NP-40 in PIPES [piperazine-N,N'-bis 2-ethanesulfonic acid] buffer (100 mM PIPES [pH 6.8], 1 mM $MgCl_2$) for 5 min and washed twice in PIPES buffer, and cytoskeletons were fixed with 1% paraformaldehyde (PFA) or methanol. Fixed trypanosomes were incubated with single or combination antibodies, anti-cMyc (monoclonal antibody IgG1 9E10, 1:20) to detect expressed INb, anti-$Tb$BILBO1 (amino acids 1-110, rabbit, 1517028 bleed 1, 1:4,000), anti-$Tb$MORN1 (rabbit 1:5,000), anti-$Tb$PFR2 (rabbit 1:200), and anti-$Tb$BLD10 (rabbit 1:2,000) for 60 min, washed twice in 1$\times$ PBS, and then incubated with secondary antibodies: anti-mouse IgG fluorescein isothiocyanate (FITC)-conjugated (Jackson, 115-095-164) and anti-rabbit Alexa fluor 594-conjugated (Invitrogen, A-11012). Kinetoplasts and nuclei were stained with 10 $\mu$l DAPI (4',6-diamidino-2-phenylindole; 10 $\mu$g/ml) in 1$\times$ PBS for 2 min, and slides were mounted with Slowfade Gold antifade reagent (Invitrogen, Molecular Probes). Images were acquired on a Zeiss Axioimager Z1 using MetaMorph software and a Roper CoolSNAP HQ2 camera. For Nbs used as probes in IFA of fixed trypanosomes, PCF Tb427 29.13 wild type (WT) and $Tb$427 29.13 pLew100-$Tb$BILBO1-3cMyc were fixed (as described above)

and incubated with Nb (0.25 mg/ml) for 1 h, followed by anti-HA (Biolegend, 901513, mouse) diluted 1:200. For colabeling with $Tb$BILBO1, anti-$Tb$BILBO1 (1-110) was used diluted 1:4,000 in 1× PBS. Secondary antibodies were used with 1:100 dilution: anti-mouse FITC-conjugated (Sigma, F2012), anti-rabbit Alexa fluor 594-conjugated (Invitrogen, A11012). For IFA on U-2 OS cells, cells were fixed and processed as described in references 23 and 25. The primary antibodies anti-$Tb$BILBO1 (1-110), 1:4,000 dilution, anti-$Tb$BILBO1 (IgM) mouse monoclonal, 5F2B3 (13), 1:2 dilution, anti-HA epitope tag (mouse monoclonal, Biolegend, 1:1,000), and anti-HA (rabbit, GeneTex GTX115044, 1:1,000) were incubated for 1 h in a dark, moist chamber. After two 1× PBS washes, cells were incubated for 1 h, respectively, with the secondary antibodies goat anti-rabbit IgG (H+L) conjugated to Alexa fluor 594 (Molecular Probes A-11012, 1:400), goat anti-mouse IgM conjugated to Alexa fluor 594 (Invitrogen, 1:400), goat anti-mouse whole molecule conjugated to FITC (Sigma F-2012, 1:400), and goat anti-rabbit IgG (H+L) conjugated to Alexa fluor 594 (Molecular Probes A-11012, 1:400). Purified Nb48$_{::HA::6His}$ raised against $Tb$BILBO1 was used as a probe (10 $\mu$g/ml) and incubated for 1 h in a dark, moist chamber. After two PBS washes, cells were incubated for 1 h with the secondary antibody directed against the HA tag, anti-HA epitope tag (Biolegend, 1:1,000), and then washed twice in PBS and incubated 1 h with the tertiary antibody goat anti-mouse whole molecule conjugated to FITC (Sigma, 1:400). The intrabody anti-green fluorescent protein (GFP) nanobody 3× HA-tagged (INbGFP$_{::3HA}$) was probed with anti-HA mouse IgG Biolegend, diluted 1:1,000 in PBS fetal calf serum (FCS) 10%, saponin 0.1%, and probed with anti-mouse goat whole molecule FITC, Sigma, diluted 1:100 in PBS FCS 10%, saponin 0.1%. The nuclei were stained with DAPI (0.25 $\mu$g/ml in PBS for 5 min). Slides were washed and mounted with Prolong (Molecular Probes S-36930). Images were acquired on a Zeiss Imager Z1 microscope with Zeiss 63× objective (NA 1.4) using a Photometrics Coolsnap HQ2 camera and Metamorph software (Molecular Devices) and processed with ImageJ (NIH).

**Flagella preparation for IFA.** Ten milliliters of mid-log PCF cells were collected and washed once in 1× PBS for 5 min at 1,000 × $g$ at RT. Cells were extracted with 1 ml 1% NP-40 in 100 mM PIPES (pH 6.9), 1 mM MgCl$_2$ for 7 min at RT to make cytoskeletons. Cytoskeletons were centrifuged for 10 min at 5,000 × $g$ at 4°C and further extracted for 15 to 30 min on ice with 1% NP-40 in 100 mM PIPES buffer plus 2mM MgCl$_2$, containing 1 M KCl. Flagella were centrifuged for 5 min at 5,000 × $g$ at 4°C, the supernatant was removed, and flagella were resuspended in 500 $\mu$l PIPES buffer. Flagella were washed twice in 100 mM PIPES buffer plus 1 mM MgCl$_2$ for 5 min at 5,000 × $g$ at 4°C before being deposed on poly-lysine coated slides and left to adhere for 5 to 10 min. The flagella were fixed at −20°C in MeOH or in 3% PFA as described above.

**Transmission electron microscopy.** Ten milliliters of mid-log-phase procyclic, $Tb$427 29.13 wild-type (WT) cells, pLew100-Nb48-3cMyc (induced for 24 h) was fixed, dehydrated, and embedded, cut, and stained as in reference 12, with the exception that tannic acid was omitted and cells were initially fixed in their culture medium by the addition of glutaraldehyde to a final concentration of 2.5% and paraformaldehyde at 3.7% for 2 h and then pelleted at 1,000 × $g$ and fixed for 2 h in 2.5% glutaraldehyde in phosphate buffer (pH 7.4).

**Immuno-electron microscopy.** Ten milliliters of mid-log-phase ~5 × 10$^6$/ml pLew100-Nb48-3cMyc cells was induced with 1 $\mu$g/ml tetracycline for 24 h. Five milliliters of induced cells was harvested at 1,000 × $g$ for 5 min, washed twice with PBS by centrifugation (1,000 × $g$), and resuspended in 500 $\mu$l of PBS. Freshly glow-discharged, Formvar and carbon-coated G200-Ni grids (EMS) were floated on the droplets, and the cells were allowed to adhere for 15 min. Grids were then moved onto a drop of 1% NP-40 in PEME buffer (100 mM PIPES [pH 6.8], 1 mM MgCl$_2$, 0.1 mM EGTA) (5 min, RT), followed by incubation on a 500 $\mu$l drop of 1% NP-40, 1 M KCl in PEME buffer for 15 min (3 × 5 min, 4°C). Flagella were washed twice (2 × 5 min) in PEME buffer at RT, equilibrated, and blocked on 50 $\mu$l drops of 2% fish skin gelatin (Sigma-Aldrich G7041) or 0.5% BSA, 0.1% Tween 20 in PBS, incubated in 25 $\mu$l of primary antibody diluted in blocking buffer (each primary antibody was used either alone or mixed with a second primary antibody), 1 h at room temperature (RT). Anti-cMyc IgG mouse monoclonal antibody (9E10, purified, a kind gift from Klaus Ersfeld) was diluted 1:10 or protein G purified/concentrated supernatant (diluted 1:50). Anti-$Tb$BILBO1 (1-110) was diluted 1:200 or 1:400. Grids were blocked and incubated in secondary antibody for 1 h at RT (anti-mouse goat GMTA 5 nm gold, and/or anti-rabbit goat 15 nm gold GAR15, BBInternational) 1:10 in 0.2% fish skin gelatin in PBS. Grids were blocked and fixed in 2.5% glutaraldehyde in 0.2% fish skin gelatin in PBS. Samples were negatively stained with 1% aurothioglucose (Sigma-Aldrich) (10 $\mu$l for 30 s). Micrographs were taken on a Phillips Technai 12 transmission electron microscope at 100 kV.

## SUPPLEMENTAL MATERIAL

Supplemental material is available online only.

**SUPPLEMENTAL FILE 1**, PDF file, 7.3 MB.

## ACKNOWLEDGMENTS

We thank our lab members for insightful comments on the manuscript, Brooke Morriswood (University of Wurzburg, Germany) for the anti-$Tb$MORN1 antibody, Keith Gull (University of Oxford, England) for the anti-TAT1 (anti-tubulin) antibody, Zyin Li (University of Texas, USA) for the anti-$Tb$BLD10 antibody, Klaus Ersfeld (University of Bayreuth, Germany) for the anti-cMyc (9E10) antibody, Frédéric Bringaud (University of Bordeaux, Bordeaux, France) for the anti-enolase antibody, and Nicolas Biteau (University of Bordeaux, Bordeaux, France) for the anti-$Tb$PFR2 rabbit polyclonal antibody. The Robinson lab acknowledges support by the Centre

National de la Recherche Scientifique (CNRS; https://www.cnrs.fr/) and the Université de Bordeaux (https://www.u-bordeaux.fr/). C.B.R. and M.R.R. were supported by fellowships from the French Ministry of Higher Education, Research and Innovation (https://www.enseignementsup-recherche.gouv.fr/). We acknowledge funding from the Fondation pour le Recherche Médicale (FRM, https://www.frm.org/) (FRM) grant number FDT202001010783 awarded to M.R.R. This work was also supported by Laboratoire d'Excellence (https://www.enseignementsup-recherche.gouv.fr/cid51355/laboratoires-d-excellence.html) through the LabEx ParaFrap (grant number ANR-11-LABX-0024) awarded to D.R.R. and ANR-FWF PRCI [ANR-20-CE91-0003] to M.B.

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
