## [Reviewer comments · Microbiology Spectrum]

Microbiology Spectrum

Intrabody induced cell death by targeting the *T. brucei* cytoskeletal protein TbBILBO1

Christine Broster-Reix, Miharisoa Ramanantsalama, Carmelo Di Primo, Laëtitia Minder, Mélanie Bonhivers, Denis Dacheux, and Derrick Robinson

Corresponding Author(s): Derrick Robinson, University of Bordeaux, CNRS, Microbiologie UMR 5234

Review Timeline:

Submission Date:	July 16, 2021
Editorial Decision:	August 16, 2021
Revision Received:	September 17, 2021
Accepted:	September 30, 2021

Editor: Vasant Muralidharan

Reviewer(s): The reviewers have opted to remain anonymous.

Transaction Report:

DOI: <https://doi.org/10.1128/Spectrum.00915-21>

August 16, 2021

Prof. Derrick R. Robinson
University of Bordeaux, CNRS, Microbiologie UMR 5234
Bordeaux, Nouvelle-Aquitaine 33076
France

Re: Spectrum00915-21 (Intrabody induced cell death by targeting the *T. brucei* cytoskeletal protein TbBILBO1)

Dear Prof. Derrick R. Robinson:

Thank you for submitting your manuscript to Microbiology Spectrum. The reviewers appreciated the well controlled data as well as the utility of nanobodies and intrabodies. However, the model describing how intrabodies lead to BILBO1 degradation was unconvincing. The data at present do not support this model and the discussion of this result should reflect that or you could add proteasome-inhibition data to provide stronger support for this model.

When submitting the revised version of your paper, please provide (1) point-by-point responses to the issues raised by the reviewers as file type "Response to Reviewers," not in your cover letter, and (2) a PDF file that indicates the changes from the original submission (by highlighting or underlining the changes) as file type "Marked Up Manuscript - For Review Only". Please use this link to submit your revised manuscript - we strongly recommend that you submit your paper within the next 60 days or reach out to me. Detailed information on submitting your revised paper are below.

Link Not Available

Sincerely,

Vasant Muralidharan

Journals Department
Reviewer comments:

Reviewer #1 (Comments for the Author):

In this work, the authors have generated a set of nanobodies against the flagellar pocket collar protein BILBO1. They test the affinity and efficacy of three nanobodies that represent the family of clones they identified in their screen and identify one, Nb48, that binds with high affinity. They show that these reagents work effectively as detection reagents for the FPC and BILBO1 in immunofluorescence, iEM, and western blots. They then test the effects of employing Nb48 as an intrabody, which allows the nanobody to sequester BILBO1 in cells and should produce a loss-of-function phenotype. Cells expressing the intrabody have significant FPC and hook complex defects, producing the FP-less flagella phenotype seen in the BILBO1 RNAi, which causes rapid lethality. The nanobody is then used in an exogenous expression system (mammalian cells), where it is able to bind to BILBO1 as a reagent applied to cells or as an intrabody. The intrabody expression also disrupts the morphology of polymers that BILBO forms in mammalian cells, suggesting that it is able to disrupt the formation of higher-order structures.

Overall, the work shows the potential of nanobodies for work in *T. brucei*, especially if they can be employed as intrabodies to rapidly disrupt protein function. I have 3 points that I think should be addressed.

1. Considering that one of the main points of this work is to show the production and testing of the nanobodies, the authors should not use a western blot in figure 1 to document the production of the nanobodies. A recombinant protein such as the nanobody should be documented using a coomassie gel to show purity of the protein. An uncropped version of the gel in the supplementary figure could work for this.

2. I am not certain that the authors have shown that the intrabody binding to BILBO1 leads to the specific degradation of BILBO1. The western blots in figure 4E show that at 48 h essentially nothing is left in the intrabody-induced cells. To me, this would argue that BILBO1 sequestration leads to such disruption of the FP that at some point you are getting cell lysis. If intrabody expression was truly causing selective depletion of BILBO1, that should be happening fairly quickly after intrabody expression, I'd expect its binding of BILBO1 to occur within minutes inside the cell, with degradation happening on the order of hours at most. The fact that we don't see BILBO1 depletion at 24 h is surprising. Is it possible that the intrabody is only able to capture free BILBO1? It may be able to bind to BILBO1 that has already been incorporated into the FPC, but not disrupt its function. This difference in effect on free vs FPC-incorporated BILBO1 could explain the delay in the phenotype. Perhaps an MG132 treatment to inhibit the proteasome could show some degradation of BILBO1 upon intrabody expression, if it's happening? Do the authors have images of what the cells look like after 2 d of intrabody expression- are they severely compromised?

3. The differences between Nb48 and Nb9 are very interesting. Considering that there is only a 2-fold difference in antibody affinity between them it is surprising that you can not detect Nb9 binding without overexpression of BILBO1, especially since both reagents bind in the low-nanomolar range. Could this difference have something to do with the location of the Nb9 epitope, and not just its affinity? Was epitope mapping performed on Nb9? Does Nb9 localize to the FPC when used as an intrabody?

More minor points:

Line 204: There is no evidence that MORN1 binds to MTs directly, change phrasing.

Fig 3B: It would be useful to put which size beads was used for each label next to the antibody names.

Figure 8: by their own convention, should the green channel be labeled INb48, because the nanobody is not actually expressed in the cells in this case, from what I can understand?

Can Nb48 kill trypanosomes if it is added to media? BSFs especially? Are they taken up by cells at all?

Reviewer #2 (Comments for the Author):

The paper by Reix et al outlines the validation of a nanobody against TbBILBO1, an important cytoskeletal protein localized at the flagellar pocket of *Trypanosoma brucei*. As the site of material entering the cell, understanding the structural features and mechanics of the flagellar pocket is important for understanding basic features of *T. brucei* biological processes. The authors illustrate that nanobodies raised against TbBILBO1 behave similarly to traditional antibodies, and further demonstrate that endogenous expression of the nanobody interferes with the function of the protein, both in the *T. brucei* system and a heterologous mammalian cell system. The validation and intrabody inhibition experiments performed using the nanobody are well controlled and nicely executed on a technical level; the data is sound and believable. The data support the notion that nanobodies might be helpful as an alternative to both traditional antibody methods and RNAi methods. One nanobody against a cytoskeletal *Trypanosoma brucei* protein has already been characterized, so this study demonstrates that the approach can work for at least one other protein in this class. I don't see any necessity for adding any experiments, but I think the paper could be framed slightly differently from the writing standpoint to make clearer why a researcher might choose a nanobody approach over an antibody or RNAi approach.

Major Points

1. The section on the use of the nanobody in a heterologous system should be condensed. Figure 8 shows that the nanobody against TbBILBO1 (Nb58) recognizes the full length BILBO1 protein in mammalian cells along with a couple of truncations. Figure 9 shows that endogenous expression of the nanobody interferes with polymer formation of full length TbBILBO1 and Figure 10 demonstrates that this is also true for 2 of the truncations. There are separate sections of the text describing these figures. However, I think it would be much easier to combine Figures 8-10 into one figure so that one can more easily compare the phenotypes between Nb58 staining and the intrabody expression of Nb58. As is the author is flipping back and forth between Figures 8, 9, and 10 as these comparisons are described. Rather than having separate sections of the text for each of these Figures they could be consolidated into one section on heterologous expression of TbBILBO1 and INb48. The mutated EFh could also be moved to the supplement to make more room for the combined figure.

2. As I was reading through this paper, I found myself thinking, wow, these intrabody assays seems like a huge amount of work while conveying information that is very similar to assays using RNAi, which on the surface seem cheaper and faster. However, the authors bring up some nice points about the limitations of RNAi in the concluding section. I think if this material were moved

to the introduction then the reader would be more motivated to consider how nanobodies could be used in their own research labs, especially if they work on an organism without RNAi.

3. In my opinion the introduction dwells a trifle too much on the therapeutic applications of nanobodies and intrabodies. The material having to do with therapeutic applications in lines 83-112 could be condensed to 1-2 sentences on the utility of these tools in a therapeutic setting. In addition, the therapeutic potential of nanobodies within the *T. brucei* system was not very clear to me. If the authors have a specific application in mind, it would be useful to outline it as an example, because as it stands the therapeutic potential within *T. brucei* comes off as a little vague. To be clear, the benefits to basic science are more than enough to justify the development of this technology without needing to invoke a therapeutic application. However, if the authors are going to spend appreciable time discussing therapeutic applications, then it would be nice to have a more clear cut description of why this would be useful in this parasite system.

4. The authors convincingly demonstrate that the nanobody produced in the study faithfully reproduces localization with traditional antibodies and can also be used for EM. They further demonstrate that previously characterized TbBILBO1 RNAi phenotypes are recapitulated using the nanobody. What I found myself wondering was whether the nanobody produced a unique or additional set of data beyond what was already known using antibody and RNAi approaches. It's perfectly ok if that is not the case, since the value of the study does not depend on this result. However, if the use of nanobodies introduced new and valuable information about this previously studied protein, then it would be great if the authors could highlight that in both the results and the discussion. The authors produced a very nice set of data using intrabodies that they stated was similar to the RNAi data, but I wanted to know if this technology allowed them to go further in the characterization of the biology of BILBO1 than they had been able to do with the RNAi system.

Minor Points

1. Line 43. Sentence should read Trypanosomes are flagellated protists comprised of pathogenic species. Same is true for line 56, should read organized cytoskeleton comprised primarily of microtubules.
2. Line 156 needs a paragraph break.
3. Line 159 The MtQ was defined earlier, though a reminder about it's function wouldn't hurt.
4. Line 189 STED should be defined.
5. Line 199 missing the word on after carried out
6. Lines 202-205 don't flow well in this paragraph and could be cut.
7. Line 207-208. Are the callouts for S. Figure 1B and 1C reversed?

Staff Comments:

Preparing Revision Guidelines

For complete guidelines on revision requirements, please see the Instructions to Authors at [link to page]. **Submissions of a paper that does not conform to Microbiology Spectrum guidelines will delay acceptance of your manuscript.**

Please return the manuscript within 60 days; if you cannot complete the modification within this time period, please contact me. If you do not wish to modify the manuscript and prefer to submit it to another journal, please notify me of your decision immediately so that the manuscript may be formally withdrawn from consideration by Microbiology Spectrum.

If you would like to submit an image for consideration as the Featured Image for an issue, please contact Spectrum staff.

The paper by Reix et al outlines the validation of a nanobody against *TbBILBO1*, an important cytoskeletal protein localized at the flagellar pocket of *Trypanosoma brucei*. As the site of material entering the cell, understanding the structural features and mechanics of the flagellar pocket is important for understanding basic features of *T. brucei* biological processes. The authors illustrate that nanobodies raised against *TbBILBO1* behave similarly to traditional antibodies, and further demonstrate that endogenous expression of the nanobody interferes with the function of the protein, both in the *T. brucei* system and a heterologous mammalian cell system. The validation and intrabody inhibition experiments performed using the nanobody are well controlled and nicely executed on a technical level; the data is sound and believable. The data support the notion that nanobodies might be helpful as an alternative to both traditional antibody methods and RNAi methods. One nanobody against a cytoskeletal *Trypanosoma brucei* protein has already been characterized, so this study demonstrates that the approach can work for at least one other protein in this class. I don't see any necessity for adding any experiments, but I think the paper could be framed slightly differently from the writing standpoint to make clearer why a researcher might choose a nanobody approach over an antibody or RNAi approach.

Major Points

1. The section on the use of the nanobody in a heterologous system should be condensed. Figure 8 shows that the nanobody against *TbBILBO1* (Nb58) recognizes the full length BILBO1 protein in mammalian cells along with a couple of truncations. Figure 9 shows that endogenous expression of the nanobody interferes with polymer formation of full length *TbBILBO1* and Figure 10 demonstrates that this is also true for 2 of the truncations. There are separate sections of the text describing these figures. However, I think it would be much easier to combine Figures 8-10 into one figure so that one can more easily compare the phenotypes between Nb58 staining and the intrabody expression of Nb58. As is the author is flipping back and forth between Figures 8, 9, and 10 as these comparisons are described. Rather than having separate sections of the text for each of these Figures they could be consolidated into one section on heterologous expression of *TbBILBO1* and INb48. The mutated EFh could also be moved to the supplement to make more room for the combined figure.
2. As I was reading through this paper, I found myself thinking, wow, these intrabody assays seems like a huge amount of work while conveying information that is very similar to assays using RNAi, which on the surface seem cheaper and faster. However, the authors bring up some nice points about the limitations of RNAi in the concluding section. I think if this material were moved to the introduction then the reader would be more motivated to consider how nanobodies could be used in their own research labs, especially if they work on an organism without RNAi.
3. In my opinion the introduction dwells a trifle too much on the therapeutic applications of nanobodies and intrabodies. The material having to do with therapeutic applications in lines 83-112 could be condensed to 1-2 sentences on the utility of these tools in a therapeutic setting. In addition, the therapeutic

potential of nanobodies within the *T. brucei* system was not very clear to me. If the authors have a specific application in mind, it would be useful to outline it as an example, because as it stands the therapeutic potential within *T. brucei* comes off as a little vague. To be clear, the benefits to basic science are more than enough to justify the development of this technology without needing to invoke a therapeutic application. However, if the authors are going to spend appreciable time discussing therapeutic applications, then it would be nice to have a more clear cut description of why this would be useful in this parasite system.

4. The authors convincingly demonstrate that the nanobody produced in the study faithfully reproduces localization with traditional antibodies and can also be used for EM. They further demonstrate that previously characterized *Tb*BILBO1 RNAi phenotypes are recapitulated using the nanobody. What I found myself wondering was whether the nanobody produced a unique or additional set of data beyond what was already known using antibody and RNAi approaches. It's perfectly ok if that is not the case, since the value of the study does not depend on this result. However, if the use of nanobodies introduced new and valuable information about this previously studied protein, then it would be great if the authors could highlight that in both the results and the discussion. The authors produced a very nice set of data using intrabodies that they stated was similar to the RNAi data, but I wanted to know if this technology allowed them to go further in the characterization of the biology of BILBO1 than they had been able to do with the RNAi system.

Minor Points

1. Line 43. Sentence should read Trypanosomes are flagellated protists **comprised** of pathogenic species. Same is true for line 56, should read organized cytoskeleton comprised primarily of microtubules.
2. Line 156 needs a paragraph break.
3. Line 159 The MtQ was defined earlier, though a reminder about it's function wouldn't hurt.
4. Line 189 STED should be defined.
5. Line 199 missing the word on after carried out
6. Lines 202-205 don't flow well in this paragraph and could be cut.
7. Line 207-208. Are the callouts for S. Figure 1B and 1C reversed?

Re: Spectrum00915-21 (Intrabody induced cell death by targeting the *T. brucei* cytoskeletal protein TbBILBO1)

Dear Prof. Derrick R. Robinson:

Thank you for submitting your manuscript to Microbiology Spectrum. The reviewers appreciated the well controlled data as well as the utility of nanobodies and intrabodies. However, the model describing how intrabodies lead to BILBO1 degradation was unconvincing. The data at present do not support this model and the discussion of this result should reflect that or you could add proteasome-inhibition data to provide stronger support for this model.

When submitting the revised version of your paper, please provide (1) point-by-point responses to the issues raised by the reviewers as file type "Response to Reviewers," not in your cover letter, and (2) a PDF file that indicates the changes from the original submission (by highlighting or underlining the changes) as file type "Marked Up Manuscript - For Review Only". Please use this link to submit your revised manuscript - we strongly recommend that you submit your paper within the next 60 days or reach out to me. Detailed information on submitting your revised paper are below.

<https://spectrum.msubmit.net/cgi-bin/main.plex?el=A2QF3Bqcz1A4DTNT6I7A9ftdesbxYIqOvQj379mShBOnIqZ>

Reviewer comments:

Reviewer #1 (Comments for the Author):

In this work, the authors have generated a set of nanobodies against the flagellar pocket collar protein BILBO1. They test the affinity and efficacy of three nanobodies that represent the family of clones they identified in their screen and identify one, Nb48, that binds with high affinity. They show that these reagents work effectively as detection reagents for the FPC and BILBO1 in immunofluorescence, iEM, and western blots. They then test the effects of employing Nb48 as an intrabody, which allows the nanobody to sequester BILBO1 in cells and should produce a loss-of-function phenotype. Cells expressing the intrabody have significant FPC and hook complex defects, producing the FP-less flagella phenotype seen in the BILBO1 RNAi, which causes rapid lethality. The nanobody is then used in an exogenous expression system (mammalian cells), where it is able to bind to BILBO1 as a reagent applied to cells or as an intrabody. The intrabody expression also disrupts the morphology of polymers that BILBO forms in mammalian cells, suggesting that it is able to disrupt the formation of higher-order structures.

Overall, the work shows the potential of nanobodies for work in *T. brucei*, especially if they can be employed as intrabodies to rapidly disrupt protein function. I have a 3 points that I think should be addressed.

1. Considering that one of the main points of this work is to show the production and testing of the nanobodies, the authors should not use a western blot in figure 1 to document the production of the nanobodies. A recombinant protein such as the nanobody should be documented using a coomassie gel to show purity of the protein. An uncropped version of the gel in the supplementary figure could work for this.

Author response (blue font)

We thank the reviewer for this comment and agree. Coomassie gels for the purification of all three nanobodies are now in Supp figure 1.

2. I am not certain that the authors have shown that the intrabody binding to BILBO1 leads to the specific degradation of BILBO1. The western blots in figure 4E show that at 48 h essentially nothing is left in the intrabody-induced cells. To me, this would argue that BILBO1 sequestration leads to such disruption of the FP that at some point you are getting cell lysis. If intrabody expression was truly causing selective depletion of BILBO1, that should be happening fairly quickly after intrabody expression, I'd expect its binding of BILBO1 to occur within minutes inside the cell, with degradation happening on the order of hours at most. The fact that we don't see BILBO1 depletion at 24 h is surprising. Is it possible that the intrabody is only able to capture free BILBO1? It may be able to bind to BILBO1 that has already been incorporated into the FPC, but not disrupt its function. This difference in effect on free vs FPC-incorporated BILBO1 could explain the delay in the phenotype.

Author response

We thank the reviewer for raising this point and, in retrospect, we agree that after 24h of INb48 induction there is no INb48 or BILBO1 protein degradation but at 48h induction there degradation probably due to cell lysis. The reduction of BILBO1 level is thus a downstream and unspecific event. Taking from the reviewer's comments and our data we now say;

After 24h of INb48 induction there is no INb48 or BILBO1 protein degradation and the BILBO1 RNAi-like phenotypes are apparent, but at 48h induction there is considerable protein degradation probably due to cell lysis. INb48 binding to BILBO1 might result in BILBO1 sequestration, which could lead to intense disruption of the FP and that at some point this induces cell lysis. It is also possible that the intrabody binds to BILBO1 that has already been incorporated into the FPC, but does not disrupt its function. If the intrabody is only able to capture free BILBO1 the difference in effect on free vs FPC-incorporated BILBO1 could explain the delay in the phenotype. In this context, it is noteworthy that we also observed dramatic changes in BILBO1 polymers when co-expressed with INb48 in U2-OS cells. The binding of INb48 to newly forming BILBO1 polymers could prevent correct FPC formation by inducing malformation of the growing BILBO1 polymers. Indeed, figures 5B and D (and Supplementary Figure S3 D and E) illustrate the formation of unusual, BILBO1 positive, linear polymers, when INb48 was expressed in trypanosomes, instead of the annular polymers normally present in a wild-type FPC. We are currently investigating the knockdown effect in more detail and also the trafficking of FPC proteins to the FPC, to ascertain whether the MtQ is implicated. This may also answer, in part, the question of how the FPC, a flagellum-associated complex of proteins, is built.

Perhaps an MG132 treatment to inhibit the proteasome could show some degradation of BILBO1 upon intrabody expression, if it's happening?

Author response

We thank the reviewer for this comment. However, we have accepted the editor's suggestion to modify the discussion to reflect a different model, which we have described above, thus omitting the need for the suggested MG132 test.

Do the authors have images of what the cells look like after 2 d of intrabody expression - are they severely compromised?

Author response

We do have images of 48hr of INb48 expression and they are now included in the Supp Fig 3D and E. These IFA images of 48hpi of INb48::3cMyc, show that cells are severely compromised and show cell cycle disruption including cells with more than two flagella.

3. The differences between Nb48 and Nb9 are very interesting. Considering that there is only a 2-fold difference in antibody affinity between them it is surprising that you can not detect Nb9 binding without overexpression of BILBO1, especially since both reagents bind in the low-nanomolar range. Could this difference have something to do with the location of the Nb9 epitope, and not just its affinity? Was epitope mapping performed on Nb9? Does Nb9 localize to the FPC when used as an intrabody?

Author response

- We thank the reviewer for this comment. Yes, we agree, it is interesting that there is a difference in affinity between Nb48 and Nb9. We believe that it is highly likely that differences in epitope binding is/(are) the main reason(s). Given that a protein epitope consists of a minimum of 6 aa, we cannot conclude that the two Nbs recognize the same sequence of 6 aa on a domain comprising 336 aa. We know the amino acid sequences of these nanobodies and it is acknowledged that Nbs bind to their epitope *via* three epitope binding regions (CDR,1,2 and 3), but the main and most important is CDR3.

The amino acid sequences of these nanobodies are currently part of a patent application (they can be provided upon request if confidentiality is guaranteed). Nb48 CDR3 contains two glutamic acids and a tryptophan where as Nb9 does not. However, the CDR3 of Nb9 contains two valines, a glutamine and a proline, whereas Nb48 does not. CDR3 of Nb48 contains 15 amino acids whereas Nb9 contains 18. Thus, the CDR3 of Nb48 is smaller and more likely to access smaller, cryptic and different epitopes compared to Nb9. This can, in part, explain a difference in detection by IFA, if we consider that the two epitopes are clearly different. This would also suggest that binding of Nb48 to its BILBO1 epitope is possible under the, *in vivo*, native, BILBO1 expression levels within the FPC, whereas binding of Nb9 is impaired due the possible presence of BILBO1 partner proteins that could partially shield the epitope or because the epitope is not fully exposed. Over-expression of BILBO1 would most probably produce and expose much more unshielded and/or accessible BILBO1 epitope allowing Nb9 to bind. This would also explain why Nb9 was initially selected using ELISA assays since the selection protocol was based on the use of purified recombinant BILBO1. A modified version of this information has been added to the discussion.

- We did do epitope mapping of Nb9 and it is now included as Figure 2F. As with Nb48, Nb9 binds to the truncation T4, domain of BILBO1. However, based on the differences in amino acid type and distribution as well as differences and CDR3 size, we suggest that Nb9 binds to a different epitope to Nb48 within the T4, C' terminal domain.

- Nb9 does indeed target the FPC when used as an intrabody, we have included an image in supporting Sup Figure 2E to indicate this.

More minor points:

Line 204: There is no evidence that MORN1 binds to MTs directly, change phrasing.

Author response

Done. The line now reads "It is important to note that Gheiratmand et al., 2013, Esson et al., 2012 and Albisetti 2017 et al., have shown that TbSPEF1 (a microtubule binding and bundling protein) and TbMORN1 are localized to the MtQ of flagella thus clearly indicating that they remain attached to this structure after harsh extraction and indicating that labelling of the MtQ is not artefactual (73, 20, 25)".

Fig 3B: It would be useful to put which size beads was used for each label next to the antibody names.

Done.

Figure 8: by their own convention, should the green channel be labeled INb48, because the nanobody is not actually expressed in the cells in this case, from what I can understand?

Thank you. This is a typographical error, and it has been rectified.

Can Nb48 kill trypanosomes if it is added to media? BSFs especially? Are they taken up by cells at all?

These are questions that we have planned to address. We agree with the reviewer about the potential use of Nb48 external to the cell. Unfortunately, we were not able to obtain large enough quantities of Nb48 to test uptake and movement within the cell, but initial trial tests suggest that Nb48, when added to the media of a BSF culture does not kill the trypanosomes. In addition, we did not expect this approach to work because the FPC is sub-pellicular and contained in the cytoplasm, whereas endocytosed molecules pass through the endo-membrane system and are recycled or targeted to the lysosome. Endocytosed Nbs would therefore remain within endocytotic or recycled vesicles and not have direct contact with the cytoplasm.

Reviewer #2 (Comments for the Author):

The paper by Reix et al outlines the validation of a nanobody against TbBILBO1, an important cytoskeletal protein localized at the flagellar pocket of *Trypanosoma brucei*. As the site of material entering the cell, understanding the structural features and mechanics of the flagellar pocket is important for understanding basic features of *T. brucei* biological processes. The authors illustrate that nanobodies raised against TbBILBO1 behave similarly to traditional antibodies, and further demonstrate that endogenous expression of the nanobody interferes with the function of the protein, both in the *T. brucei* system and a heterologous mammalian cell system. The validation and intrabody inhibition experiments performed using the nanobody are well controlled and nicely executed on a technical level; the data is sound and believable. The data support the notion that nanobodies might be helpful as an alternative to both traditional antibody methods and RNAi methods. One nanobody against a cytoskeletal *Trypanosoma brucei* protein has already been characterized, so this study demonstrates that the approach can work for at least one other protein in this class. I don't see any necessity for adding any experiments, but I think the paper could be framed slightly differently from the writing standpoint to make clearer why a researcher might choose a nanobody approach over an antibody or RNAi approach.

Major Points

1. The section on the use of the nanobody in a heterologous system should be condensed. Figure 8 shows that the nanobody against TbBILBO1 (Nb58) recognizes the full length BILBO1 protein in mammalian cells along with a couple of truncations. Figure 9 shows that endogenous expression of the nanobody interferes with polymer formation of full length TbBILBO1 and Figure 10 demonstrates that this is also true for 2 of the truncations. There are separate sections of the text describing these figures. However, I think it would be much easier to combine Figures 8-10 into one figure so that one can more easily compare the phenotypes between Nb58 staining and the intrabody expression of Nb58. As is the author is flipping back and forth between Figures 8, 9, and 10 as these comparisons are described. Rather than having separate sections of the text for each of these Figures they could be consolidated into one section on heterologous expression of TbBILBO1 and INb48. The mutated EFh could also be moved to the supplement to make more room for the combined figure.

Author response

We thank the reviewer for helping to make the manuscript easier to follow. We have combined figures 8, 9, and 10 to make a combined figure and have kept the EF-hand mutation data because it does not take up space in terms of figure width and it allows all the data to be visualized in one figure.

2. As I was reading through this paper, I found myself thinking, wow, these intrabody assays seems like a huge amount of work while conveying information that is very similar to assays using RNAi, which on the surface seem cheaper and faster. However, the authors bring up some nice points about the limitations of RNAi in the concluding section. I think if this material were moved to the introduction then the reader would be more motivated to consider how nanobodies could be used in their own research labs, especially if they work on an organism without RNAi.

Done.

3. In my opinion the introduction dwells a trifle too much on the therapeutic applications of nanobodies and intrabodies. The material having to do with therapeutic applications in lines 83-112 could be condensed to 1-2 sentences on the utility of these tools in a therapeutic setting. In addition, the therapeutic potential of nanobodies within the *T. brucei* system was not very clear to me. If the authors have a specific application in mind, it would be useful to outline it as an example, because as it stands the therapeutic potential within *T. brucei* comes off as a little vague. To be clear, the benefits to basic science are more than enough to justify the development of this technology without needing to invoke a therapeutic application. However, if the authors are going to spend appreciable time discussing therapeutic applications, then it would be nice to have a more clear cut description of why this would be useful in this parasite system.

We agree with the reviewer that lines 83-112 maybe too long. We have substantially reduced it, but feel that it is worth mentioning most of the major points outlined in the original manuscript, because they describe the origin of nanobodies, the fact that nanobodies are being used globally in many areas of biology and human medicine, which ultimately provides an insight that reaches a wider audience.

4. The authors convincingly demonstrate that the nanobody produced in the study faithfully reproduces localization with traditional antibodies and can also be used for EM. They further demonstrate that previously characterized TbBILBO1 RNAi phenotypes are recapitulated using the nanobody. What I found myself wondering was whether the nanobody produced a unique or additional set of data beyond what was already known using antibody and RNAi approaches. It's perfectly ok if that is not the case, since the value of the study does not depend on this result. However, if the use of nanobodies introduced new and valuable information about this previously studied protein, then it would be great if the authors could highlight that in both the results and the discussion.

The authors produced a very nice set of data using intrabodies that they stated was similar to the RNAi data, but I wanted to know if this technology allowed them to go further in the characterization of the biology of BILBO1 than they had been able to do with the RNAi system.

Author response

Currently, we do not have much more information about the use of nanobodies with BILBO1 because this work was based on a proof of concept idea, but we do note that ; "In conclusion, we induced a BILBO1 RNAi phenotype not by knocking down protein expression but rather by preventing protein function. This indicates that an intrabody-induced, protein binding approach can prevent the formation of a new FPC most likely even in presence of BILBO1 interacting proteins. This would indeed suggest that BILBO1 is the main protein performer in FPC biogenesis". We will be looking very carefully at the potential data that can be obtained from further studies with nanobodies.

Minor Points

1. Line 43. Sentence should read Trypanosomes are flagellated protists comprised of pathogenic species. Same is true for line 56, should read organized cytoskeleton comprised primarily of microtubules. Done.

2. Line 156 needs a paragraph break. Done.

3. Line 159 The MtQ was defined earlier, though a reminder about it's function wouldn't hurt. Done.

4. Line 189 STED should be defined. Done.

5. Line 199 missing the word on after carried out Done.

6. Lines 202-205 don't flow well in this paragraph and could be cut. We have modified the sentence to put it into context. From previous experiences, our data showing labelling of the MtQ was considered artefactual. The use of this sentence is to inform

readers that proteins remain attached to the MtQ after salt and detergent extraction and that our labelling, is not artefactual.

7. Line 207-208. Are the callouts for S. Figure 1B and 1C reversed? Yes. Thank you for noticing this. We have now fixed the typo.

September 30, 2021

Prof. Derrick R. Robinson
University of Bordeaux, CNRS, Microbiologie UMR 5234
Bordeaux, Nouvelle-Aquitaine 33076
France

Re: Spectrum00915-21R1 (Intrabody induced cell death by targeting the *T. brucei* cytoskeletal protein TbBILBO1)

Dear Prof. Derrick R. Robinson:

Congratulations, your manuscript has been accepted, and I am forwarding it to the ASM Journals Department for publication. You will be notified when your proofs are ready to be viewed.

Sincerely,

Vasant Muralidharan
Editor, Microbiology Spectrum

Journals Department
Supplemental Material FOR Publication: Accept